# Neural Networks as Paths through the Space of Representations

## Abstract

Deep neural networks implement a sequence of layer-by-layer operations that are each relatively easy to understand, but the resulting overall computation is generally difficult to understand. We consider a simple hypothesis for interpreting the layer-by-layer construction of useful representations: perhaps the role of each layer is to reformat information to reduce the "distance" to the desired outputs. With this framework, the layer-wise computation implemented by a deep neural network can be viewed as a path through a high-dimensional representation space. We formalize this intuitive idea of a "path" by leveraging recent advances in *metric* representational similarity. We extend existing representational distance methods by computing geodesics, angles, and projections of representations, going beyond mere layer distances. We then demonstrate these tools by visualizing and comparing the paths taken by ResNet and VGG architectures on CIFAR-10. We conclude by sketching additional ways that this kind of representational geometry can be used to understand and interpret network training, and to describe novel kinds of similarities between different models.

**Keywords:** representational similarity, metric spaces, geometry, neural network expressivity, visualization

## 1 Introduction

A core design principle of modern neural networks is that they process information serially, progressively transforming inputs until the information is in a format that is immediately usable for some task (Rumelhart et al., 1988; LeCun et al., 2015). This idea of composing sets of simple units to construct more complicated functions is central to both artificial neural networks and how neuroscientists conceptualize various functions in the brain (Kriegeskorte, 2015; Richards et al., 2019; Barrett et al., 2019).

Our work is motivated by a spatial analogy for information-processing: we imagine that outputs are "far" from inputs if the mapping between them is complex, or "close" if it is simple. In this spatial analogy, any one layer of a neural network contributes a single step, and the composition of many steps transports representations along a path towards the desired target representation. Formalizing this intuition requires a method to quantify if any two representations are "close" (similar) or "far" (dissimilar) (Kriegeskorte, 2009; Kornblith et al., 2019). In order to use this kind of spatial or geometric analogy for neural representations, we need some way to quantify the "distance" between representations. We build on recent work introducing *metrics* for quantifying representational dissimilarity (Williams et al., 2021; Shahbazi et al., 2021). Representational dissimilarity is quantified using a function $d(\mathbf{X}, \mathbf{Y}) : \mathbb{X} \times \mathbb{X} \to \mathbb{R}^+$ that takes in two matrices of neural data and outputs a nonnegative value for their dissimilarity. Here, $\mathbb{X} = \bigcup_{n=1,2,3,...} \mathbb{R}^{m \times n}$ is the space of all $m \times n$ matrices for all $n$. The matrices $\mathbf{X}$ and $\mathbf{Y}$ could be, for instance, the values of two hidden layers in a network with $n_x$ and $n_y$ units, respectively, in response to $m$ inputs.

What are desirable properties of such a representational dissimilarity function? Previous work has argued that any sensible dissimilarity function should be nonnegative, so $d(\mathbf{X}, \mathbf{Y}) \geq 0$, and should return zero between any **equivalent** representations, so $d(\mathbf{X}, \mathbf{Y}) = 0 \Leftrightarrow \mathbf{X} \sim \mathbf{Y}$, where $\mathbf{X} \sim \mathbf{Y}$ means that $\mathbf{X}$ and $\mathbf{Y}$ are in the same equivalence class. For example, we may wish to design the function d so that $d(\mathbf{X}, \mathbf{Y}) = 0$ if $\mathbf{Y}$ is a shifted copy of $\mathbf{X}$, or if it is a non-degenerate scaling,

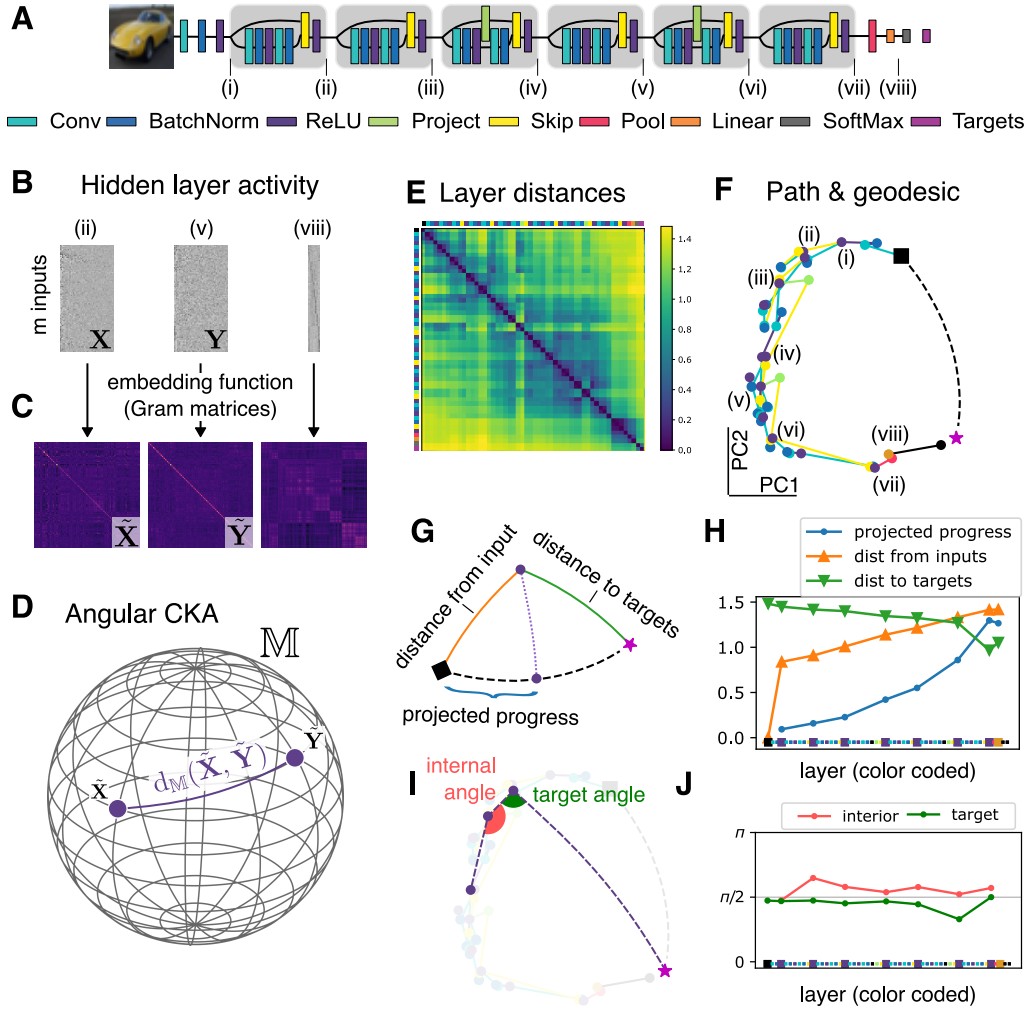

Figure 1: Representation paths of Resnet-14 model trained on CIFAR-10, evaluated using the Angular CKA metric with a linear kernel. **A)** Schematic of Resnet-14 architecture with color-coded layers. Gray boxes correspond to residual blocks. **B, C, D)** Matrices of neural data are converted into points in the metric space of Angular CKA. The outputs of each block form a $m \times n$ matrix for $m$ examples and $n$ channels **(B)**; data from layers (ii), (v), and (vii) in panel A are shown. Outputs are transformed into centered and normalized Gram matrices by the embedding function **(C)**. The resulting points are on a spherical manifold where the arc distance between points provides a measure of the distance between representations **(D)**. **E)** Pairwise distance between layers computed using Angular CKA. **F)** 2D embedding of the network's path using multi-dimensional scaling (MDS) down to 15D followed by PCA. Includes points for the input pixels (black square), target class labels (purple star), and points calculated from the geodesic between input and labels (black dashed line). **G)** Three types of distances plotted in **(H)**: distance of layer from input (orange ▲), distance to target (green ▼), and projected distance along the geodesic from input to target (blue). **I)** Two types of angles plotted in **(J)**: "internal angle" or the angle between adjacent path segments (red), and "target angle" or the angle between each segment and the geodesic from segment to targets (green). Note that we treat each residual block as a single step or segment of the path.

rotation, or affine transformation of $\mathbf{X}$ (Kornblith et al., 2019; Williams et al., 2021; Shahbazi et al., 2021). A second desirable property is that d is **symmetric**, so $\mathrm{d}(\mathbf{X}, \mathbf{Y}) = \mathrm{d}(\mathbf{Y}, \mathbf{X})$. A third is that d satisfies the **triangle inequality**, or $\mathrm{d}(\mathbf{X}, \mathbf{Y}) \le \mathrm{d}(\mathbf{X}, \mathbf{Z}) + \mathrm{d}(\mathbf{Z}, \mathbf{Y})$. As argued by Williams et al. (2021), a representational dissimilarity function that fails to satisfy the triangle inequality can lead

to errant results when, for instance, clustering or embedding representations based on their pairwise dissimilarity. A dissimilarity function that satisfies all of the above properties – equivalence, symmetry, and the triangle inequality – qualifies as a **metric**[1] on $\mathbb{X}$ (Burago et al., 2001). Examples of metrics between neural representations were recently developed independently by Williams et al. (2021) and Shahbazi et al. (2021).

We are interested in using metrics between neural representations to explore how representations evolve *spatially* as they are transformed through the hidden layers of deep networks. Not all metrics are sufficient for the kind of spatial reasoning – that is, not all metrics can be interpreted as *distances*. For example, consider the trivial metric

$$\mathrm{d}(\mathbf{X}, \mathbf{Y}) = \begin{cases} 0 & \text{if } \mathbf{X} \sim \mathbf{Y} \\ 1 & \text{otherwise} \end{cases}.$$

This is a valid metric according to the equivalence, symmetry, and triangle inequality criteria, but it is useless as a tool for characterizing distances. To be interpretable as a measure of distance, $\mathrm{d}(\mathbf{X}, \mathbf{Y})$ must satisfy an intuitive fourth condition called **rectifiability**: the distance between any two points must be realizable as the (infimum of the) sum of distances of segments along a path between them (Burago et al., 2001). While not all metrics are rectifiable (such as the trivial metric above), it is perhaps unsurprising that this condition is met by many sensible metrics, including those already developed by Williams et al. (2021) and Shahbazi et al. (2021). In fact, all metrics considered in this paper are **Riemannian metrics**, which not only implies that they are rectifiable, but further requires that points live on a smooth manifold (Burago et al., 2001). Each restriction on the type of metric comes with additional structure that we can use to inspect and visualize neural representations: rectifiability allows us to smoothly *interpolate* neural representations along a geodesic as well as compute projections and angles, and Riemannian structure allows us to meaningfully compare the *direction* of steps taken by different layers.

The main contributions of this paper are

- We put forward the spatial "path" analogy for deep neural networks, and quantitatively evaluate it using recently developed methods for representational distance.
- We develop a toolbox for analyzing geometric properties of sequences of neural network layers, and show how existing representational distance measures imply a rich set of geometric concepts beyond mere pairwise distance.
- We create novel visualizations of how representations are transformed through the layers of deep networks.
- We apply these techniques to compare paths taken by wide and deep residual networks, as well as four VGG architectures, all trained on CIFAR-10, finding differences in both the magnitude and direction of steps taken by layers in wide versus deep models Nguyen et al. (2021).

## 2  PRELIMINARIES

### 2.1  RELATED WORK

One motivation for thinking of neural networks as paths is that it provides a compelling analogy for the way that complex functions (deep networks) can be composed out of simple parts (layers). Indeed, it is well known that both deeper (Poole et al., 2016; Raghu et al., 2017; Rolnick and Tegmark, 2017) and wider (Hornik et al., 1989) neural network architectures can express a larger class of functions than their shallower or narrower counterparts. However, much less is known about how implementing a *particular* complex function constrains the role of individual layers and intermediate representations in the intervening layers between input and output. Our work is in line with other recent efforts to characterize the features learned in hidden layers as smoothly varying between inputs and outputs (Chan et al., 2020; Yang et al., 2022; He and Su, 2022). In our path framework,

---

[1]It would be more precise to say it is a "metric" on the quotient space of the equivalence class $\mathbf{X} \sim \mathbf{Y}$, or a "pseudometric" on $\mathbb{X}$, but we suppress this distinction throughout to avoid excess verbiage. For a more thorough treatment of equivalence classes, see (Williams et al., 2021).

relatively narrow layers take shorter steps, and chaining them together increases the total achievable length, while wide layers are capable of taking a single large step, and all such steps contribute by moving "closer" to the targets. Our work is a first step, so to speak, towards formalizing this notion of composing simple functions in *geometric* terms.

There is a rich literature applying geometric concepts like distance between representations to formalize notions of "similarity" in neuroscience and psychology (Edelman, 1998; Jäkel et al., 2008; Rodriguez and Granger, 2017; Hénaff et al., 2019; Kriegeskorte and Wei, 2021). However, there is a crucial difference between measuring similarity or distance between points in a given space, and *measuring distances between representational spaces themselves*. The former includes questions like, "how far apart are the activation vectors for two inputs in a given layer?" The latter, which is the subject of this paper, asks instead, "how far apart are two layers' representations, considering all inputs?"

Our work is most closely related to, and draws much inspiration from recent advances in representational similarity analysis (RSA). In particular, Kornblith et al. (2019) showed that a kernel method for testing statistical dependence, known as CKA (Gretton et al., 2005; Cortes et al., 2012), is closely related to classic RSA Kriegeskorte (2009). In follow-up work, Nguyen et al. (2021) used CKA to make layer-by-layer comparisons between wide and deep networks – which we elaborate on in section 3.2 below. Independently, both Williams et al. (2021) and Shahbazi et al. (2021) developed methods to compute metrics between neural representations. Shahbazi et al. (2021) proposed using the so-called **Affine-Invariant Riemannian Metric** on the space of symmetric positive-definite matrices (Pennec, 2006; 2019). Williams et al. (2021) derived a metric variation of CKA which we call **Angular CKA**, as well as a family of **Generalized Shape Metrics**. We extend this prior work by computing not just pairwise distances, but by also introducing a suite of tools for analyzing the geometry of representation-space induced by each of these distance functions. Finally, whereas Williams et al. (2021) and Shahbazi et al. (2021) compare representations *across models*, we compare representations *within a single model* to study the transformation of information from inputs to outputs through hidden layers.

## 2.2 DISTANCE METRICS BETWEEN NEURAL REPRESENTATIONS

In our notation, a deep neural network processes an input $\mathbf{x}^0$ through a sequence of hidden layers, $\{\mathbf{x}^1, \ldots, \mathbf{x}^{L-1}\}$, and produces outputs $\mathbf{x}^L$, trained to match some target outputs (LeCun et al., 2015). In the common example of image classification, $\mathbf{x}^0$ is the input image, targets are the class label, and $\mathbf{x}^L$ a set of (log) class probabilities. We will use subscripts to index inputs, so $\mathbf{x}_i^l$ is the $n_l$-dimensional response of the $l$th hidden layer to the $i$th input, $\mathbf{x}_i^0$, with $i \in \{1, \ldots, m\}$. We use capital letters like $\mathbf{X}^l$, $\mathbf{X}^k$, or $\mathbf{Y}$ to refer to $m \times n$ matrices of neural representations in response to $m$ inputs[2], and we treat target labels as one-hot vectors (i.e. if $\mathbf{Y}$ is the target representation, then it is a $m \times 10$ matrix for the CIFAR-10 dataset (Krizhevsky, 2009)). We will adopt the convention that $l < k$ if $\mathbf{x}^l$ is a direct ancestor of $\mathbf{x}^k$ through the layers of a network (layers will in general be partially but not strictly ordered). We are interested in functions $\mathrm{d}(\mathbf{X}, \mathbf{Y})$ for that quantify the "dissimilarity" between two neural representations; when $\mathrm{d}(\mathbf{X}, \mathbf{Y})$ satisfies the four conditions of a length metric – equivalence, symmetry, triangle inequality, and rectifiability – we will say that it measures their "representational distance." Note that $\mathbf{X}$ and $\mathbf{Y}$ may be different layers of the same model or layers from different models, as long as they are evaluated on the same inputs. We use $\mathrm{d}(\mathbf{X}, \mathbf{Y})$ to refer to the distance function between arbitrary representations $\mathbf{X}$ and $\mathbf{Y}$.

We can unify and organize existing representational distance metrics by recognizing that distance metrics use a two-stage approach to defining $\mathrm{d}(\mathbf{X}, \mathbf{Y})$: first, $\mathbf{X}$ and $\mathbf{Y}$ are mapped to a common space $\mathbb{M}$ through an **embedding function** $\mathrm{f} : \mathbb{X} \to \mathbb{M}$, then distance is computed using a distance metric defined on $\mathbb{M}$. More precisely,

$$\mathrm{d}(\mathbf{X}, \mathbf{Y}) \equiv \mathrm{d}_{\mathbb{M}}(\tilde{\mathbf{X}}, \tilde{\mathbf{Y}}), \tag{1}$$

where $\tilde{\mathbf{X}} = \mathrm{f}(\mathbf{X})$ is the result of mapping $\mathbf{X}$ from $\mathbb{X}$ to $\mathbb{M}$. In all cases we consider here, $\mathbb{M}$ is a Riemannian manifold with metric $\mathrm{d}_{\mathbb{M}}$. Equivalence relations on $\mathbb{X}$ can be built into this two-stage approach in either stage: $\mathrm{d}(\mathbf{X}, \mathbf{Y})$ can be made invariant to changes in the scale of $\mathbf{X}$ either by imposing a canonical scale in $\mathrm{f}$, or by preserving scale in $\mathrm{f}$ but using a scale-invariant metric $\mathrm{d}_{\mathbb{M}}$.

---

[2]We assume convolutional layers are flattened, in which case $n$ is the product of height, width, and feature dimensions.

We implemented three existing families of representational distance, each of which can be understood as different choices for f, $\mathbb{M}$, and $d_{\mathbb{M}}$, and extended them to further compute various geometric quantities including geodesics, projections, tangent vectors, and angles. The representational distance metrics we investigate here are **Angular CKA** (with a linear kernel), **Shape Metrics** (with angular distance), and the **Affine Invariant Riemannian Metric** (with a squared exponential kernel and ridge regularization) (Williams et al., 2021; Shahbazi et al., 2021). We have found that results using Angular CKA are the most interpretable among the metrics we have tested, so our main results are presented using Angular CKA. Results using other metrics are shown in Figure C.1. In the following section, we will give mathematical details for Angular CKA; Table A.1 and Appendix A provide mathematical details on all metrics in one place, including a discussion of their invariances.

## 2.3 THE GEOMETRY OF ANGULAR CKA

Angular CKA was introduced by Williams et al. (2021) (eq (60) in their supplement). It is defined as the arccosine of CKA, which is itself derived from the Hilbert-Schmidt independence criterion (HSIC) (Gretton et al., 2005; Cortes et al., 2012; Kornblith et al., 2019). Because HSIC and CKA measure statistical dependence, distance measured by Angular CKA is small when the rows of $\mathbf{X}$ and $\mathbf{Y}$ are strongly statistically dependent, and large when they are independent. While Angular CKA was originally introduced simply as a method for computing a metric between neural representations, here we exploit the fact that Angular CKA is the arc-length on a Hypersphere to compute additional geometric properties of the space.

Angular CKA is equivalent to arc-length distance on the spherical manifold consisting of *centered* and *normalized* $m \times m$ Gram matrices. Let $\boldsymbol{G}_{\mathbf{X}}$ denote the Gram matrix of $\mathbf{X}$, i.e. $\boldsymbol{G}_{\mathbf{X}ij}$ is given by the inner-product between the $i$th and $j$th rows of $\mathbf{X}$. Optionally, this inner-product may be computed using a kernel; following previous work (Kornblith et al., 2019; Nguyen et al., 2021), results in this paper use Linear CKA, i.e. we use $\boldsymbol{G}_{\mathbf{X}} = \mathbf{X}\mathbf{X}^{\top}$. Other kernels are in principle admissible and available in our python package at [link redacted]. The *normalized* and *centered* Gram matrix is given by

$$\tilde{G}_{\mathbf{X}} = \frac{H G_{\mathbf{X}} H}{||H G_{\mathbf{X}} H||_{\mathrm{F}}}$$

where $\boldsymbol{H} = \mathbb{I}_m - \frac{1}{m}\mathbf{1}\mathbf{1}^{\top}$ is the $m \times m$ **centering matrix**, and $||\cdot||_{\mathrm{F}}$ is the Frobenius norm of a matrix. The Riemannian manifold $\mathbb{M}$ for Angular CKA consists of all centered and normalized symmetric positive definite matrices; it is a **sphere** because $\left\langle \tilde{G}_{\mathbf{X}}, \tilde{G}_{\mathbf{X}} \right\rangle_{\mathrm{F}} = 1$, where $\langle \boldsymbol{A}, \boldsymbol{B} \rangle_{\mathrm{F}} = \mathrm{Tr}(\boldsymbol{A}^{\top}\boldsymbol{B})$ is the Frobenius inner-product.

Comparing to equation (1), we can see that the **embedding function** for Angular CKA is $f(\mathbf{X}) = \tilde{G}_{\mathbf{X}}$. Distance according to Angular CKA is

$$\begin{aligned} d(\mathbf{X}, \mathbf{Y}) &= d_{\mathbb{M}}(f(\mathbf{X}), f(\mathbf{Y})) \\ &= d_{\mathbb{M}}(\tilde{G}_{\mathbf{X}}, \tilde{G}_{\mathbf{Y}}) \\ &= \arccos\left( \left\langle \tilde{G}_{\mathbf{X}}, \tilde{G}_{\mathbf{Y}} \right\rangle_{\mathrm{F}} \right) \end{aligned} \quad (2)$$

(Figure 1B-D).

Because Angular CKA is an arc length on a hypersphere, we can easily compute its **geodesics**: the shortest path connecting $\tilde{G}_{\mathbf{X}}$ to $\tilde{G}_{\mathbf{Y}}$ is

$$\text{geodesic}(\tilde{G}_{\mathbf{X}}, \tilde{G}_{\mathbf{Y}}, t) = \frac{\sin((1-t)\Omega)}{\sin(\Omega)}\tilde{G}_{\mathbf{X}} + \frac{\sin(t\Omega)}{\sin(\Omega)}\tilde{G}_{\mathbf{Y}}, \quad (3)$$

where $t \in [0, 1]$ is the fraction of distance along the geodesic from $\mathbf{X}$ to $\mathbf{Y}$, and $\Omega = d_{\mathbb{M}}(\tilde{G}_{\mathbf{X}}, \tilde{G}_{\mathbf{Y}})$. This is a direct application of the SLERP formula[3] for hyperspheres, applied to $\tilde{G}$-space.

For all metrics, we use numerical optimization methods to project a point in $\mathbb{M}$ onto the geodesic spanning two other points. The projection of $\tilde{\mathbf{X}}$ onto the geodesic spanning $\tilde{\mathbf{Y}}$ and $\tilde{\mathbf{Z}}$ can be found

---

[3]https://en.wikipedia.org/wiki/Slerp

by minimizing the distance from $\tilde{\mathbf{X}}$ to $\texttt{geodesic}(\tilde{\mathbf{Y}}, \tilde{\mathbf{Z}}, t)$ with respect to $t$. In practice, we solve this as an optimization problem with respect to $t$. Figure 1G demonstrates one way that this idea of projecting neural representations can be used: we project each hidden layer onto the geodesic connecting inputs (raw pixels) to targets (one-hot labels). (Note that we keep only the representations labeled (i) through (vii) in Figure 1A so that the resulting path consists only of steps that perform comparable operations). This provides a more gradual picture of the "progress" being made by each layer towards the targets than mere distance between layers and inputs or targets (Figure 1H).

For all metrics, we can compute angles between any three neural representations using the **tangent space** defined by the metric. The **logarithmic map** is a function that takes in two points in $\mathbb{M}$ and returns a tangent vector that "points" from one to the other (Burago et al., 2001). The logarithmic map for Angular CKA from $\tilde{G}_{\mathbf{X}}$ to $\tilde{G}_{\mathbf{Y}}$ is

$$\log_{\tilde{G}_{\mathbf{X}}}\left(\tilde{G}_{\mathbf{Y}}\right) = \boldsymbol{W} \arccos\left(\left\langle \tilde{G}_{\mathbf{X}}, \tilde{G}_{\mathbf{Y}} \right\rangle_{\mathrm{F}}\right) \tag{4}$$

where $\boldsymbol{W}$ is the unit tangent vector at $\tilde{G}_{\mathbf{X}}$ pointing towards $\tilde{G}_{\mathbf{Y}}$, given by

$$\boldsymbol{W} = \frac{\tilde{G}_{\mathbf{Y}} - \tilde{G}_{\mathbf{X}} \left\langle \tilde{G}_{\mathbf{X}}, \tilde{G}_{\mathbf{Y}} \right\rangle_{\mathrm{F}}}{||\tilde{G}_{\mathbf{Y}} - \tilde{G}_{\mathbf{X}} \left\langle \tilde{G}_{\mathbf{X}}, \tilde{G}_{\mathbf{Y}} \right\rangle_{\mathrm{F}} ||_{\mathrm{F}}} .$$

We then compute **angles** between triplets of points by computing the inner-product of their tangent vectors. In the case of Angular CKA in particular, let $\boldsymbol{W}_{\mathbf{XY}}$ denote the tangent vector pointing from $\mathbf{X}$ to $\mathbf{Y}$, i.e. the result of $\log_{\tilde{G}_{\mathbf{X}}}(\tilde{G}_{\mathbf{Y}})$. Then,

$$\theta(\tilde{G}_{\mathbf{X}}, \tilde{G}_{\mathbf{Y}}, \tilde{G}_{\mathbf{Z}}) = \arccos\left(\frac{\langle \boldsymbol{W}_{\mathbf{YX}} \boldsymbol{W}_{\mathbf{YZ}} \rangle_{\mathrm{F}}}{||\boldsymbol{W}_{\mathbf{YX}}||_{\mathrm{F}} ||\boldsymbol{W}_{\mathbf{YZ}}||_{\mathrm{F}}}\right) \tag{5}$$

is the angle of the $\mathbf{XYZ}$ triangle at $\mathbf{Y}$. Equations (3)–(5) are simply applications of the geometry of a hypersphere to the space of $m \times m$ centered and normalized Gram matrices. Figure 1I demonstrates two types of angles that are of interest: "internal angles," which quantify how straight a path is, and "target angles," which quantify to what extent each step points in the direction of the targets. Interestingly, we find that nearly all internal angles are orthogonal, and only later layers begin to take steps in the direction of the targets (Figure 1J).

## 2.4 COMPARISONS WITH OTHER METRICS

The main results in this paper are presented using Angular CKA for ease of exposition. Mathematical details for other metrics can be found in Table A.1 and Appendix A. Figure C.1 gives results like those in Figure 1 using the metrics proposed by Williams et al. (2021) (Angular Shape and Euclidean Shape) and Shahbazi et al. (2021) (Affine-Invariant Riemannian).

## 3 EXPERIMENTS

### 3.1 HOW PATHS CHANGE OVER TRAINING

Whereas Figure 1 visualizes properties of the path taken by a well-trained Resnet-14 model, here we ask how properties of its path emerge over training. For this, we trained a new model of the same architecture, and logged model parameters every 10 batches. Since the vast majority of changes occur in the first epoch, Figure 2 shows how distances, projected distances, and angles change for every 10 steps in the first epoch, plus a single additional point after 20 epochs to show where values converge after longer training.

At initialization, the distance from inputs initially increases (Figure 2B), but the distance to targets is constant (Figure 2A). This implies that the network's representations are changing in some direction that is orthogonal to making "progress" towards the targets. Indeed, we find that plotting projected progress is a useful way to visualize training (Figure 2C). At initialization, projected progress is zero for all layers, and it quickly begins to ramp upwards after a few training steps.

We hypothesized that training would cause the network's path to become more *straight*, but surprisingly we found that this is not the case. In fact, internal angles begin slightly above $\pi/2$ at initialization, and concentrate towards $\pi/2$ as training progresses (Figure 2D). This means that the function

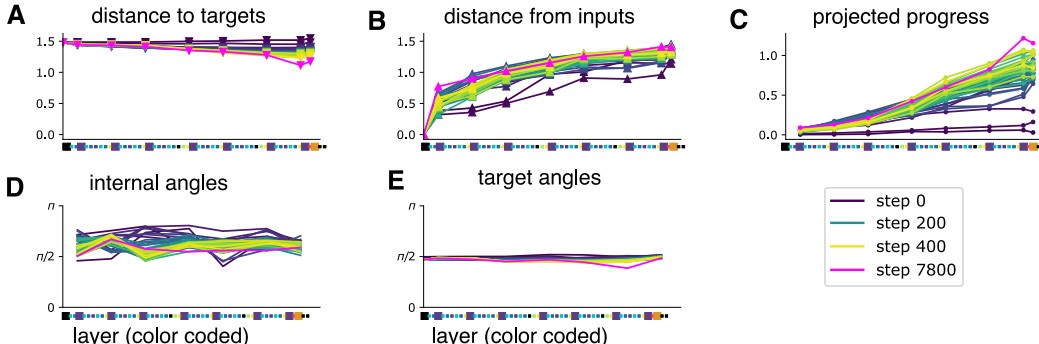

Figure 2: Evolution of paths over training for a Resnet-14 model trained on CIFAR-10. We plot lines every 10 batches for the first epoch (blue to yellow), and an additional line at epoch 20 (magenta). The first row plots the same distances as in Figure 1**H**. **A)** Distance to targets (one-hot labels) from hidden layers as a function of layer depth. **B)** Distance from inputs (pixels) to hidden layers. **C)** Projected progress, or the distance from inputs to the projection of hidden layers onto the geodesic connecting inputs to targets. **D)** Internal angles, or angles between adjacent path segments versus increasing layer depth. **E)** Target angles, or angles between each segment and the geodesic from its starting point to the targets.

implemented by each residual block applies an orthogonal operation (in representation space) compared to the blocks before or after it. To our knowledge, this kind of orthogonality between layers has not been previously observed.

Figure 2E shows how "target angles" evolve over training. Consistent with past work, this visualization shows that the emergence of layers that point in the direction of the targets does not happen until later in training.

## 3.2 COMPARING PATHS OF WIDE VERSUS DEEP MODELS

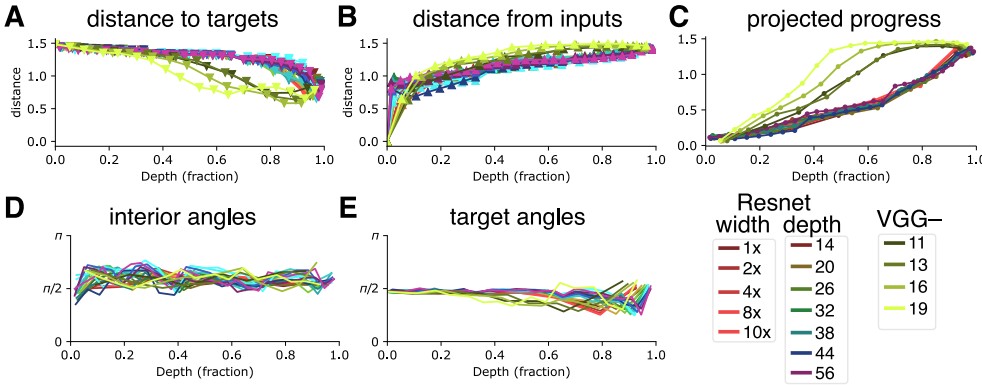

Figure 3: Comparison between paths of different model architectures. We compare residual models by width (brightness) and by depth (hue), as well as VGG networks (yellows). Plots follow layout of figure 2: **A)** distance to targets, **B)** distance from inputs, **C)** projected distance to targets, **D)** angle between path segments, and **E)** angle between segment and target.

Inspired by the results of Nguyen et al. (2021), we compare paths taken by "wide" versus "deep" models. In Figure 3, we show the same quantities we computed in Figure 2, but comparing different Resnet widths and depths. Our results confirm the main findings of Nguyen et al. (2021), namely that wide and deep Resnets compute similar functions — in other words, wide-model paths and deep-model paths all follow roughly the same trajectory in representational space. Note that the x-axis for these plots is scaled by the depth of each model; the fact that all Resnets are superimposed

on each other means that all of them distribute the function of each layer relatively evenly over their depth. This is not universally true of all models – we also compare with VGG, and see salient differences between the VGG and Resnet architectures, where the progress made in VGG (without skip connections) is much more dependent on absolute rather than relative depth.

## 3.3 Decomposing each step into "progress" and "deviation"

We next asked to what extent each step in the path, i.e. each residual block, moves representations towards the targets (**progress**) or in orthogonal directions (**deviation**; Figure 4A). Note that unlike "projected progress" above, which was measured relative to the input-targets geodesic, here we compare each step from $\mathbf{X}^l$ to $\mathbf{X}^{l+1}$ to the geodesic connecting *current* position $\mathbf{X}^l$ to the targets. A model with D blocks provides D-1 measures of both progress and deviation – one for each block. Figure 4B summarizes the net contribution of progress and deviation for all layers in models with varying architectures. Note the following salient effects: first, there is a strong overall upward trend – for every block that makes 0.1 radians of progress towards the targets (recall that Angular CKA is an arc-length), it incurs some deviation of about 0.3 radians in an orthogonal direction, and this trend is remarkably uniform for both Resnet and VGG architectures. Second, deeper models are lower and to the left; deeper models parcel out both progress and deviation among their layers. Third, we can identify an effect of width at constant depth; at a constant depth, wider models incur less deviation than their narrower counterparts.

## 4 Discussion

In this work, we interpret neural networks spatially as paths from inputs to outputs through a space of intermediate representations. These paths have rich geometric structure, inherited from computing distances between representations, and we can use that structure to build intuitions about network structure, training, and comparisons between models.

We investigated three families of representational distance metrics — Angular CKA, Shape Metrics, and the Affine Invariant metric on symmetric positive definite matrices (Williams et al., 2021; Shahbazi et al., 2021). Surprisingly, we found that trained networks take circuitous paths according to all of these metrics, and deviate far from the shortest paths from inputs to targets which are defined geometrically by each representational distance metric. There are three potential explanations for this. First, networks may be taking short paths according to some metric other than those we investigated here, implying that our metrics may not reflect the distance between representations in terms of ease of computation. Second, neural networks may fail to take efficient paths. The distance metrics we consider are all differentiable, and so an interesting question for future work is whether networks can be regularized to take shorter paths, and whether such regularization will improve or reduce their performance or generalization ability. Third, it could be that networks take the shortest and most direct path which is possible under some architectural constraints, which may prevent the hidden layers of the network from moving directly along the geodesic. This explanation must be at least partially true, since the dimensionality of the representation space $\mathbb{M}$ generally exceeds the number of parameters in each layer/block of the network.

We were also surprised to discover that according to all metrics we investigated, network paths tended to be *jagged*, and consist predominantly of 90° angle turns. Although it is well known that random directions in high-dimensional spaces such as the representation space $\mathbb{M}$ tend to be nearly always orthogonal, this does not explain why we found that path angles are straighter at initialization and become more orthogonal during training. This is surprising because the training objective ostensibly should encourage all layers to point in the same direction towards the targets. This means that steps become *more orthogonal* through training. This is an especially surprising result in light of recent work by Chan et al. (2020) that suggests that a sequence of residual blocks can be interpreted as a sequence of small gradient steps optimizing a *rate reduction* objective; in this interpretation, all layers ought to be moving in the same general direction. However, this is not what we find in our models trained by backpropagation and where angles are quantified using Angular CKA (nor using other metrics – see Figure C.1). Ultimately, ours is an empirical finding which suggests that future theoretical work is needed to interpret the *direction* of steps taken in representation space in the context of a given representational distance metric, and to understand which directions are realizable by a given network architecture.

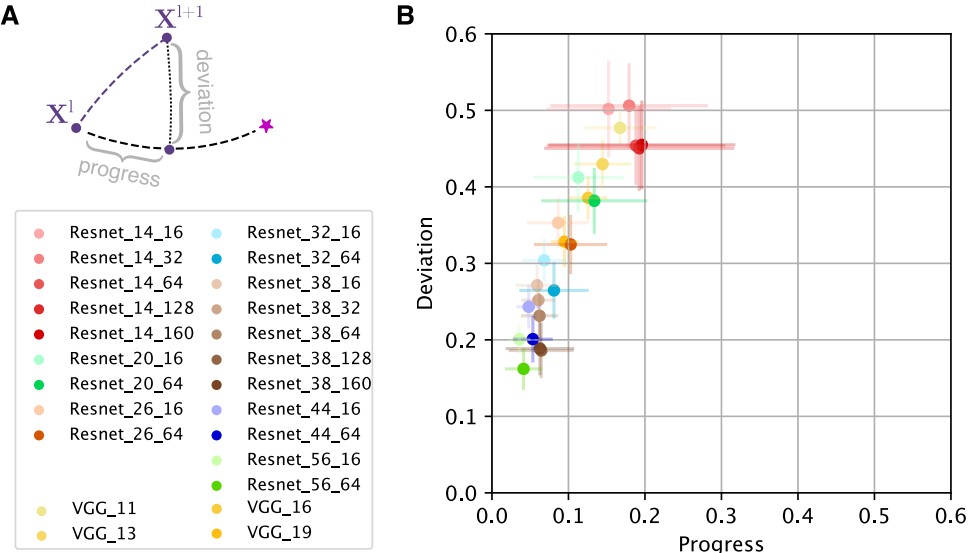

Figure 4: **A)** For each model, we decompose every path segment into "progress" and "deviation" components, where "progress" is the component along the geodesic from the segment start to the targets, and "deviation" is the component which is orthogonal to the geodesic. **B)** plots the average (circle) and $\pm$ standard error (horizontal and vertical lines) "progress" and "deviation" over all layers of a variety of model architectures. Note that the names of Resnets indicate depth followed by width, and the names of VGG models indicate depth.

As described in the introduction, we are motivated to develop a general spatial analogy for information-processing, where complex transformations of representations cover more "distance" than simple ones. In this sense, a measure of representational distance ought to reflect the *function complexity* of transforming $\mathbf{X}$ into $\mathbf{Y}$. We chose to extend and compare existing representational distance metrics in order to build directly on previous work, but the metrics we evaluated here may not be interpretable as measures of function complexity. An exciting avenue for future work is thus to derive a measure of representational distance directly from a measure of function complexity. Such a measure will likely violate the axioms of a distance metric. For example, the complexity of a function and its inverse are in general not equal, and so it may be desirable to have $d(\mathbf{X}, \mathbf{Y}) < d(\mathbf{Y}, \mathbf{X})$ if the transformation from $\mathbf{X}$ to $\mathbf{Y}$ can be implemented by a simpler function than the inverse. Notably, many of the geometric properties studied here (shortest paths, angles, projection, etc.) can be extended to the asymmetric case using the theory of Finsler rather than Riemannian manifolds (Burago et al., 2001).

The analogy of neural networks as paths in a representation space brings together ideas about representational similarity and the expressivity of deep networks, marrying these techniques with intuitive and mathematically rigorous geometric concepts. Our work takes a first step in exploring the possibilities of this new geometric framework, and we anticipate that it will spark new insights about model design, model training, and model comparison.

## 5 REPRODUCIBILITY STATEMENT

Our model specifications, training recipe, and data augmentation are all based on standard research practice: we use default training options for all models taken from the OpenLTH framework[4]. Our standalone PyTorch library for metric representational distance is available at [redacted github link], and our repository to train and analyze models specific to this paper is available at [redacted github link].

---

[4]https://github.com/facebookresearch/open_lth

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

# A  DETAILED INFORMATION ON METRICS

| Distance Metric | Manfiold $\mathbb{M}$ | Distance $\mathrm{d}_{\mathbb{M}}(\tilde{\mathbf{X}}, \tilde{\mathbf{Y}})$ |
|:---:|:---:|:---:|
| Angular CKA$^{\dagger}$ | $Gram_m$ | $\arccos\left(\frac{\langle\tilde{\mathbf{X}},\tilde{\mathbf{Y}}\rangle_{\mathrm{F}}}{\sqrt{\langle\tilde{\mathbf{X}},\tilde{\mathbf{X}}\rangle_{\mathrm{F}}\langle\tilde{\mathbf{Y}},\tilde{\mathbf{Y}}\rangle_{\mathrm{F}}}}\right)$ |
| Angular Shape$^{\dagger}$ | $\mathbb{R}^{m\times p}$ | $\min_{\mathbf{R}}\arccos\left(\frac{\langle\tilde{\mathbf{X}},\tilde{\mathbf{Y}}\mathbf{R}\rangle_{\mathrm{F}}}{\sqrt{\langle\tilde{\mathbf{X}},\tilde{\mathbf{X}}\rangle_{\mathrm{F}}\langle\tilde{\mathbf{Y}},\tilde{\mathbf{Y}}\rangle_{\mathrm{F}}}}\right)$ |
| Euclidean Shape$^{\dagger}$ | $\mathbb{R}^{m\times p}$ | $\min_{\mathbf{R}}\frac{1}{m}\sum_{i=1}^{m}\|\|\tilde{\mathbf{X}}_i - \tilde{\mathbf{Y}}_i\mathbf{R}\|\|_2$ |
| Affine Invariant Riemannian | $Sym_m^+$ or $Sym_p^+$ | $\sum_i\log(d_i)^2$ where $d_i$ is the $i$th eigenvalue of $\tilde{\mathbf{X}}^{-\frac{1}{2}}\tilde{\mathbf{Y}}\tilde{\mathbf{X}}^{-\frac{1}{2}}$ |

Table A.1: Summary of representational distance metrics. $Gram_m$ is the set of $m \times m$ centered Gram matrices. $Sym_m^+$ is the set of $m \times m$ symmetric positive definite matrices. $Dist_m$ is the set of $m \times m$ pairwise-distance matrices. $p$ is a hyperparameter used to set the dimensionality of points in $\mathbb{M}$. $\|\|\mathbf{X}\|\|_{\mathrm{F}}$ denotes the Frobenius norm, and $\langle\mathbf{X},\mathbf{Y}\rangle_{\mathrm{F}}$ denotes the Frobenius inner-product. $\mathbf{R}$ is a $p\times p$ orthonormal matrix, found by solving the orthogonal Procrustes problem to maximally align $\tilde{\mathbf{X}}$ and $\tilde{\mathbf{Y}}$. Metrics marked with "$\dagger$" are due to Williams et al. (2021). The Affine Invariant Riemannian metric is due to Pennec (2006; 2019); Shahbazi et al. (2021). For further details on each metric and its implied geometry, see Appendix A.

As summarized in equation (1), all distance metrics between neural representations operate in two stages: first, a layer's activity $\mathbf{X} \in \mathbb{X}$ is transformed into a point in some canonical space $\mathbb{M}$ through an **embedding function** f, and second distances are measured in that shared space. In the following subsections we give details for each metric in Table A.1.

We say a metric is **scale-invariant** if $\mathrm{d}(\mathbf{X}, \alpha\mathbf{X}) = 0$ for all scalars $\alpha \neq 0$. A metric is **shift-invariant** if $\mathrm{d}(\mathbf{X}, \mathbf{X}+\mathbf{1}\mathbf{b}^{\top}) = 0$ for any $n-$dimensional vector $\mathbf{b}$. A metric is **rotation-invariant** if $\mathrm{d}(\mathbf{X}, \mathbf{X}\mathbf{R}) = 0$ for any $n \times n$ orthonormal matrix $\mathbf{R}$. A metric is **affine-invariant** if $\mathrm{d}(\mathbf{X}, \mathbf{X}\mathbf{A}) = 0$ for any full-rank $n \times n$ matrix $\mathbf{A}$.

For all geometric calculations, we drew a subset of $m = 1000$ items randomly from the test set, and used the same subset throughout. We verified that $m = 1000$ was sufficient to reliably estimate the geometric quantities of interest in both the AngularCKA and Angular Shape metrics , by randomly resampling $m = 1000$ points from the test set multiple times and inspecting the variance of computed quantities (Figure C.3). Targets (class labels) are converted to 1-hot vectors before embedding.

Our Python implementation of various quantities from Riemannian geometry draws much inspiration from the `geomstats` Python package (Miolane et al., 2020). Our analyses in the main paper were done using

- Angular CKA with $m = 1000$ and the linear kernel $k(\mathbf{x}_i, \mathbf{x}_j) = \mathbf{x}_i^{\top}\mathbf{x}_j$.
- The Angular Shape metric with $m = 1000$, $p = 100$, $\alpha = 0$.
- The Gram-matrix version of the Affine-Invariant Riemannian metric with $m = 1000$, $\epsilon = 0.1$, and a squared exponential kernel with length scale set to the median Euclidean distance of $\mathbf{X}$.

We begin with an introduction to Angular CKA, where we also review some key terms from Riemannian geometry.

## A.1  ANGULAR CKA

Angular CKA was introduced by Williams et al. (2021) (eq (60) in their supplement). It is defined as the arccosine of centered kernel alignment (CKA), which is itself derived from the Hilbert-Schmidt independence criterion (HSIC) (Gretton et al., 2005; Cortes et al., 2012; Kornblith et al., 2019).

Because HSIC and CKA measure statistical dependence, distance measured by Angular CKA is high when the rows of $\mathbf{X}$ and $\mathbf{Y}$ are statistically independent, and low when they are highly correlated.

Angular CKA is equivalent to arc-length distance on the spherical manifold consisting of *centered* and *normalized* $m \times m$ Gram matrices. Let $\boldsymbol{G_X}$ denote the Gram matrix of $\mathbf{X}$, i.e. $\boldsymbol{G_{X}}_{ij}$ is given by the inner-product between the $i$th and $j$th rows of $\mathbf{X}$. Optionally, this inner-product may be computed using a kernel; following previous work (Kornblith et al., 2019; Nguyen et al., 2021), results in this paper use Linear CKA, i.e. we use $\boldsymbol{G_X} = \mathbf{X}\mathbf{X}^\top$, but our python package supports the use of other kernels. The *normalized* and *centered* Gram matrix is given by

$$\tilde{\boldsymbol{G}}_\mathbf{X} = \frac{\boldsymbol{HG_XH}}{||\boldsymbol{HG_XH}||_\text{F}}$$

where $\boldsymbol{H} = \mathbb{I}_m - \frac{1}{m}\mathbf{1}\mathbf{1}^\top$ is the $m \times m$ **centering matrix**, and $||\cdot||_\text{F}$ is the Frobenius norm of a matrix. Note that both $\boldsymbol{G}$ and $\tilde{\boldsymbol{G}}$ are symmetric and positive definite matrices. The Riemannian manifold $\mathbb{M}$ for Angular CKA consists of all such centered, normalized, symmetric positive definite matrices; it is a **sphere** in sense that $\left\langle \tilde{\boldsymbol{G}}_\mathbf{X}, \tilde{\boldsymbol{G}}_\mathbf{X} \right\rangle_\text{F} = 1$, where $\langle \boldsymbol{A}, \boldsymbol{B} \rangle_\text{F} = \text{Tr}(\boldsymbol{A}^\top \boldsymbol{B})$ is the Frobenius inner-product.

The **embedding function** for Angular CKA is $\text{f}(\mathbf{X}) = \tilde{\boldsymbol{G}}_\mathbf{X}$. Distance according to Angular CKA is equal to arc length on the sphere consisting of centered and normalized Gram matrices:

$$\begin{aligned}
\text{d}(\mathbf{X}, \mathbf{Y}) &= \text{d}_\mathbb{M}(\text{f}(\mathbf{X}), \text{f}(\mathbf{Y})) \\
&= \text{d}_\mathbb{M}(\tilde{\boldsymbol{G}}_\mathbf{X}, \tilde{\boldsymbol{G}}_\mathbf{Y}) \\
&= \arccos\left(\left\langle \tilde{\boldsymbol{G}}_\mathbf{X}, \tilde{\boldsymbol{G}}_\mathbf{Y} \right\rangle_\text{F}\right) \, .
\end{aligned} \tag{A.1}$$

Because Angular CKA is an arc length, its geodesics lie along great circles $\tilde{\boldsymbol{G}}$-space. We therefore can compute points along the geodesic in closed-form using the SLERP formula:[5]

$$\text{geodesic}(\tilde{\boldsymbol{G}}_\mathbf{X}, \tilde{\boldsymbol{G}}_\mathbf{Y}, t) = \frac{\sin((1-t)\Omega)}{\sin(\Omega)}\tilde{\boldsymbol{G}}_\mathbf{X} + \frac{\sin(t\Omega)}{\sin(\Omega)}\tilde{\boldsymbol{G}}_\mathbf{Y} \, , \tag{(3) restated}$$

where $t \in [0, 1]$ is the fraction of distance along the geodesic from $\mathbf{X}$ to $\mathbf{Y}$, and $\Omega = \text{d}_\mathbb{M}(\tilde{\boldsymbol{G}}_\mathbf{X}, \tilde{\boldsymbol{G}}_\mathbf{Y})$.

The **tangent space** for Angular CKA is the space of all symmetric $m \times m$ matrices, and the inner-product in the tangent space is simply the Frobenius inner-product. The **logarithmic map** computes tangent vectors from a base point that point towards another point. In the case of Angular CKA, the logarithmic map from $\tilde{\boldsymbol{G}}_\mathbf{X}$ to $\tilde{\boldsymbol{G}}_\mathbf{Y}$ is a tangent vector (symmetric matrix) at $\tilde{\boldsymbol{G}}_\mathbf{X}$ given by

$$\log_{\tilde{\boldsymbol{G}}_\mathbf{X}}\left(\tilde{\boldsymbol{G}}_\mathbf{Y}\right) = \boldsymbol{W} \arccos\left(\left\langle \tilde{\boldsymbol{G}}_\mathbf{X}, \tilde{\boldsymbol{G}}_\mathbf{Y} \right\rangle_\text{F}\right) \tag{(4) restated}$$

where $\boldsymbol{W}$ is the unit tangent vector at $\tilde{\boldsymbol{G}}_\mathbf{X}$ pointing towards $\tilde{\boldsymbol{G}}_\mathbf{Y}$, given by

$$\boldsymbol{W} = \frac{\tilde{\boldsymbol{G}}_\mathbf{Y} - \tilde{\boldsymbol{G}}_\mathbf{X}\left\langle \tilde{\boldsymbol{G}}_\mathbf{X}, \tilde{\boldsymbol{G}}_\mathbf{Y} \right\rangle_\text{F}}{||\tilde{\boldsymbol{G}}_\mathbf{Y} - \tilde{\boldsymbol{G}}_\mathbf{X}\left\langle \tilde{\boldsymbol{G}}_\mathbf{X}, \tilde{\boldsymbol{G}}_\mathbf{Y} \right\rangle_\text{F}||_\text{F}} \, .$$

The **exponential map** is the inverse of the logarithmic map – it is a function that "extrapolates" a tangent vector $\boldsymbol{W}$ from a point to give another point on the manifold. In the case of Angular CKA, the exponential map is given by

$$\exp_{\tilde{\boldsymbol{G}}_\mathbf{X}}(\boldsymbol{W}) = \cos\left(||\boldsymbol{W}||_\text{F}\right)\tilde{\boldsymbol{G}}_\mathbf{X} + \text{sinc}\left(||\boldsymbol{W}||_\text{F}\right)\boldsymbol{W} \tag{A.2}$$

where $\text{sinc}(x) = \frac{\sin(x)}{x}$.

For all metrics, we compute **angles** between triplets of points by computing the inner-product of their tangent vectors. In the case of Angular CKA in particular, let $\boldsymbol{W_{XY}}$ denote the tangent vector pointing from $\mathbf{X}$ to $\mathbf{Y}$, i.e. the result of $\log_{\tilde{\boldsymbol{G}}_\mathbf{X}}(\tilde{\boldsymbol{G}}_\mathbf{Y})$. Then,

$$\theta(\tilde{\boldsymbol{G}}_\mathbf{X}, \tilde{\boldsymbol{G}}_\mathbf{Y}, \tilde{\boldsymbol{G}}_\mathbf{Z}) = \arccos\left(\frac{\langle \boldsymbol{W_{YX}}\boldsymbol{W_{YZ}} \rangle_\text{F}}{||\boldsymbol{W_{YX}}||_\text{F}||\boldsymbol{W_{YZ}}||_\text{F}}\right) \tag{(5) restated}$$

is the angle of the $\mathbf{XYZ}$ triangle.

---

[5]https://en.wikipedia.org/wiki/Slerp

### A.1.1 INVARIANCES OF ANGULAR CKA

The invariances of Angular CKA depend on the kernel used to compute the Gram matrix. In the simplest case of **Linear CKA**, the Gram matrix is simply $\boldsymbol{G_X} = \mathbf{XX}^\top$. The resulting metric is

- **shift-invariant** due to centering the Gram matrix.
- **scale-invariant** due to normalizing the Gram matrix.
- **rotation-invariant** since $(\mathbf{X}\boldsymbol{R})(\mathbf{X}\boldsymbol{R})^\top = \mathbf{X}\boldsymbol{R}\boldsymbol{R}^\top\mathbf{X}^\top = \mathbf{XX}^\top$ for any orthonormal $\boldsymbol{R}$.

However, Angular CKA with is not invariant to arbitrary affine transformations – a feature it inherits from CKA and has been argued to be an important feature of CKA (Kornblith et al., 2019). Note that when using a nonlinear kernel to compute the Gram matrix, the resulting metric may lose these invariances. However, Angular CKA with a nonlinear kernel may still be shift-, scale-, and rotation-invariant if the kernel itself has those invariances. For example the squared exponential kernel

$$\boldsymbol{G}_{ij} = k(\mathbf{x}_i, \mathbf{x}_j) = \exp\left(||\mathbf{x}_i - \mathbf{x}_j||_2^2/\tau^2\right) \tag{A.3}$$

is naturally shift- and rotation-invariant, and it can me made further scale-invariant by setting the length scale $\tau$ automatically based on the scale of the data.

## A.2 SHAPE METRICS

Williams et al. (2021) proposed using a generalization of Procrustes distance and Kendall's shape space to measure metric distance between neural representations. Shape-space and Procrustes distance are a well-studied case of a Riemannian manifold between point clouds (Nava-Yazdani et al., 2020). Williams et al. (2021) consider two different shape metrics – one *angular* shape metric and one *Euclidean* shape metric. The key idea behind both of these metrics is as follows: $m \times n$ matrices of neural data are first transformed into a common $m \times p$ space, and interpreted as a point cloud consisting of $p-$dimensional points. Then, any two point clouds are scaled and rotated so that they maximally align with each other. The final distance is then computed as some measure of discrepancy between these maximally-aligned point clouds. The behavior of these shape metrics is tuned using two hyperparameters: the dimensionality $p$, and a partial whitening parameter $\alpha$.

The role of the **embedding function** for shape metrics is to convert $n-$dimensional neural data into a canonical zero-mean $p-$dimensional space (i.e. $\mathbb{M} = \mathbb{R}^{m \times p}$ is the space of all $m \times p$ matrices whose column means are all zero). Williams et al. (2021) also include a partial whitening stage as part of the embedding. This space of $m \times p$ zero-mean (and sometimes scaled) matrices is called the **pre shape space** (Nava-Yazdani et al., 2020). In the case where $n < p$, this conversion from $n$ to $p$ dimensions is done by simply padding $\mathbf{X}$ with $p - n$ columns of all zeros. In the case where $p < n$, we reduce the dimensionality of $\mathbf{X}$ by keeping only the top $p$ principal components. Formally, let $\bar{\mathbf{X}} = \mathbf{X} - \frac{1}{m}\sum_{i=1}^m \mathbf{X}_i$ be the matrix of neural data with its mean subtracted, then

$$\tilde{\mathbf{X}} = \mathrm{f}(\mathbf{X}) = \begin{cases} \mathrm{whiten}([\bar{\mathbf{X}}, \mathbf{0}], \alpha) & \text{if } n \leq p \\ \mathrm{whiten}(\bar{\mathbf{X}}\boldsymbol{U}_{:p}, \alpha) & \text{if } n > p \end{cases} \tag{A.4}$$

where $\boldsymbol{U}_{:p}$ stands for the first $p$ principal components of $\bar{\mathbf{X}}$, as unit column vectors, and $\mathbf{0}$ is a $m \times (p - n)$ matrix of all zeros. The partial whitening function begins by computing the eigen-decomposition of its input, $m^{-1}\bar{\mathbf{X}}^\top\bar{\mathbf{X}} = \boldsymbol{V}\boldsymbol{\Sigma}\boldsymbol{V}^\top$ (here, $\boldsymbol{V}$ is a $p \times p$ orthonormal matrix containing the top principal components of $\bar{\mathbf{X}}$, and $\boldsymbol{\Sigma}$ is a diagonal matrix of variances). Then, the partial whitening stage is

$$\mathrm{whiten}(\bar{\mathbf{X}}, \alpha) = \bar{\mathbf{X}}\boldsymbol{V}\left(\alpha\mathbb{I}_p + (1 - \alpha)\boldsymbol{\Sigma}^{-\frac{1}{2}}\right)\boldsymbol{V}^\top.$$

Note that when $\alpha = 0$, this is equivalent to ZCA whitening, and when $\alpha = 1$ it leaves $\bar{\mathbf{X}}$ unchanged. All shape metric results we report are with $p = 100$ and $\alpha = 0$. We use $\alpha = 0$ because this entails further invariances, making metrics more interpretable across disparate layer shapes and types (see section A.2.1 below for details on shape metric invariances).

Both the angular and Euclidean shape metrics require *aligning* by rotating the embedded points by minimizing $||\tilde{\mathbf{X}} - \tilde{\mathbf{Y}}\boldsymbol{R}||_\mathrm{F}$ where $\boldsymbol{R}$ is a $p \times p$ orthonormal matrix. This is known as the orthogonal Procrustes problem, and its solution is given by

$$\boldsymbol{R} = \boldsymbol{V}^\top\boldsymbol{U}^\top$$

where $\boldsymbol{U}\boldsymbol{\Sigma}\boldsymbol{V}^\top = \tilde{\mathbf{X}}^\top\tilde{\mathbf{Y}}$ is a singular value decomposition of $\tilde{\mathbf{X}}^\top\tilde{\mathbf{Y}}$. The **generalized** shape metrics introduced by Williams et al. (2021) include further restrictions on $\mathbf{R}$, such as considering rotations across channel but not spatial dimensions of convolutional layers, but we omit these restrictions in our work.

In the case of **angular** shape metrics, distance is defined as

$$\mathrm{d}_{\mathbb{M}}(\tilde{\mathbf{X}}, \tilde{\mathbf{Y}}) = \arccos\left(\frac{\left\langle\tilde{\mathbf{X}}, \tilde{\mathbf{Y}}\boldsymbol{R}\right\rangle_{\mathrm{F}}}{||\tilde{\mathbf{X}}||_{\mathrm{F}}||\tilde{\mathbf{Y}}||_{\mathrm{F}}}\right). \tag{A.5}$$

In the case of **Euclidean** shape metrics, distance is defined as

$$\mathrm{d}_{\mathbb{M}}(\tilde{\mathbf{X}}, \tilde{\mathbf{Y}}) = \frac{1}{m}\sum_{i=1}^m ||\tilde{\mathbf{X}}_i - \tilde{\mathbf{Y}}_i\boldsymbol{R}||. \tag{A.6}$$

We compute **geodesics** in shape space after finding $\boldsymbol{R}$ to align $\tilde{\mathbf{Y}}$ to $\tilde{\mathbf{X}}$. Then, the geodesic from $\tilde{\mathbf{X}}$ to $\tilde{\mathbf{Y}}\boldsymbol{R}$ in the **angular** case is given by the SLERP formula as in (3):

$$\mathrm{geodesic}(\tilde{\mathbf{X}}, \tilde{\mathbf{Y}}, t) = \frac{\sin((1-t)\Omega)}{\sin(\Omega)}\tilde{\mathbf{X}} + \frac{\sin(t\Omega)}{\sin(\Omega)}\tilde{\mathbf{Y}}\boldsymbol{R}, \tag{A.7}$$

where $\Omega = \mathrm{d}_{\mathbb{M}}(\tilde{\mathbf{X}}, \tilde{\mathbf{Y}})$ is the angular shape distance. Note that this means that $\mathrm{geodesic}(\tilde{\mathbf{X}}, \tilde{\mathbf{Y}}, 1)$ results in a point that is *equivalent* but not *identical* to $\tilde{\mathbf{Y}}$.

**Tangent vectors** for Euclidean shape metrics can be any $m \times p$ matrix whose column-wise mean is zero. In the case of angular shape metrics, the tangent space is further restricted to the tangent space of the hypersphere of unit-Frobenius-norm matrices (i.e. a tangent vector $\boldsymbol{W}$ at $\tilde{\mathbf{X}}$ must satisfy $\left\langle\tilde{\mathbf{X}}, \boldsymbol{W}\right\rangle_{\mathrm{F}} = 0$ in the angular case). The tangent space is further divided into so-called **horizontal** and **vertical** subspaces, where the vertical subspace captures changes to $\tilde{\mathbf{X}}$ that leave distance invariant, i.e. rotations that are removed by alignment by $\boldsymbol{R}$, and the horizontal subspace captures changes that affect the metric (Nava-Yazdani et al., 2020). The vertical component of a tangent vector $\boldsymbol{W}$ at point $\tilde{\mathbf{X}}$ is given by $\mathrm{vert}_{\tilde{\mathbf{X}}}(\boldsymbol{W}) = \tilde{\mathbf{X}}\boldsymbol{A}$, where $\boldsymbol{A} \in \mathbb{R}^{p\times p}$ is the solution to the following Sylvester equation:

$$\tilde{\mathbf{X}}^\top\tilde{\mathbf{X}}\boldsymbol{A} + \boldsymbol{A}\tilde{\mathbf{X}}^\top\tilde{\mathbf{X}} = \boldsymbol{W}^\top\tilde{\mathbf{X}} - \tilde{\mathbf{X}}^\top\boldsymbol{W}.$$

Following the example of Miolane et al. (2020), we use the `solve_sylvester` function from Scipy to compute this (Virtanen et al., 2020). The horizontal component of a tangent vector is given by simply subtracting the vertical part of $\boldsymbol{W}$:

$$\mathrm{horz}_{\tilde{\mathbf{X}}}(\boldsymbol{W}) = \boldsymbol{W} - \mathrm{vert}_{\tilde{\mathbf{X}}}(\boldsymbol{W})\frac{\langle\mathrm{vert}_{\tilde{\mathbf{X}}}(\boldsymbol{W}), \boldsymbol{W}\rangle_{\mathrm{F}}}{||\mathrm{vert}_{\tilde{\mathbf{X}}}(\boldsymbol{W})||_{\mathrm{F}}}.$$

To compute the **angle** between any triplet of representations, we use the inner-product of tangent vectors, as in (5), but using only the *horizontal* part of each tangent vector. As in Angular CKA, we compute horizontal tangent vectors from $\tilde{\mathbf{X}}$ to $\tilde{\mathbf{Y}}$ using the **logarithmic map**, which in the case of shape metrics is given by

$$\mathrm{horizontal}\ \log_{\tilde{\mathbf{X}}}(\tilde{\mathbf{Y}}) = \tilde{\mathbf{Y}}\boldsymbol{R} - \tilde{\mathbf{X}} \tag{A.8}$$

in the Euclidean case, or

$$\mathrm{horizontal}\ \log_{\tilde{\mathbf{X}}}(\tilde{\mathbf{Y}}) = \boldsymbol{W}\arccos\left(\frac{\left\langle\tilde{\mathbf{X}}, \tilde{\mathbf{Y}}\boldsymbol{R}\right\rangle_{\mathrm{F}}}{||\tilde{\mathbf{X}}||_{\mathrm{F}}||\tilde{\mathbf{Y}}||_{\mathrm{F}}}\right) \tag{A.9}$$

where $\boldsymbol{W}$ is the unit tangent vector at $\tilde{\mathbf{X}}$ pointing towards $\tilde{\mathbf{Y}}$, given by

$$\boldsymbol{W} = \frac{\tilde{\mathbf{Y}}\boldsymbol{R} - \tilde{\mathbf{X}}\left\langle\tilde{\mathbf{X}}, \tilde{\mathbf{Y}}\boldsymbol{R}\right\rangle_{\mathrm{F}}}{||\tilde{\mathbf{Y}}\boldsymbol{R} - \tilde{\mathbf{X}}\left\langle\tilde{\mathbf{X}}, \tilde{\mathbf{Y}}\boldsymbol{R}\right\rangle_{\mathrm{F}}||_{\mathrm{F}}}.$$

(Nava-Yazdani et al., 2020). As in (A.5) and (A.6), $\boldsymbol{R}$ is the rotation matrix that optimally aligns $\tilde{\mathbf{Y}}$ to $\tilde{\mathbf{X}}$.

### A.2.1  INVARIANCES OF SHAPE METRICS

The invariances of the shape metrics depend on a variety of hyperparameter settings.

- All shape metrics are **shift-invariant** because the embedding function $\tilde{\mathbf{X}} = \mathrm{f}(\mathbf{X})$ subtracts the mean.

- All shape metrics are **rotation-invariant** because of the Procrustes alignment procedure, and because rotation does not affect the principal component projection nor the zero-padding step of (A.4).

- The angular shape metric is **scale-invariant** because (A.5) divides by the norms of $\tilde{\mathbf{X}}$ and $\tilde{\mathbf{Y}}$.

- The Euclidean shape metric is not scale-invariant in general, but it is for the special case of $\alpha = 0$, since scale is removed by whitening.

- Neither angular nor Euclidean shape metrics is **affine-invariant** in general, but both can become affine-invariant for the special case of $\alpha = 0$, since full-rank affine transforms are removed by whitening as long as $n \leq p$. However, in the $n > p$ case, an affine transformation may amplify or suppress the principal components of the data, and as a result it can affect the embedding stage (A.4).

### A.3  AFFINE INVARIANT RIEMANNIAN METRIC

The Affine Invariant Riemannian (AIR) metric is a metric between symmetric positive definite (SPD) matrices, originally derived for use in image processing (Pennec, 2006; 2019), and recently it was proposed to use it as a metric between neural representations by first converting neural data into a SPD matrix (Shahbazi et al., 2021). The **embedding function** can therefore be any function that maps $m \times n$ matrices in $\mathbb{X}$ into $\mathbb{M} = Sym_k^+$ for some $k$. Shahbazi et al. (2021) considered two possibilities for the embedding stage: either using the $m \times m$ Gram matrix $\mathrm{f}(\mathbf{X}) = \boldsymbol{G}_{\mathbf{X}}$, or using the $n \times n$ data covariance matrix $\mathrm{f}(\mathbf{X}) = \mathrm{cov}(\mathbf{X}) = \frac{1}{m-1}\mathbf{X}^\top \boldsymbol{H}\mathbf{X}$. These correspond to complementary perspectives on the nature of neural representation, analogous to the difference between representational similarity analysis and pattern component analysis (Diedrichsen and Kriegeskorte, 2017).

The challenge when using the $m \times m$ Gram matrix approach is that, without further regularization, a $m \times m$ Gram matrix has rank $n$ when $n < m$, which implies that it cannot be SPD (and the metric considers all rank-deficient matrices to be infinitely far away). To address this, our toolbox implements the AIR metric between neural representations with additional regularization options. In the Gram matrix case, we regularize in two ways: first, we compute the Gram matrix using a kernel that implicitly has an infinite feature space (so that $m$ is much less than the number of features). This alleviates the rank-deficiency problem in cases where $n < m$ but rows are unique. However, when rows of $\mathbf{X}$ contain duplicates (notably, this is true for the target labels), $\boldsymbol{G}_{\mathbf{K}}$ is still rank-deficient. To address this, we include a second regularization stage where we add a small diagonal ridge with magnitude $\epsilon$. The full embedding function in the Gram matrix case is given by

$$\mathrm{f}(\mathbf{X}) = \boldsymbol{G}_{\mathbf{X}} + \epsilon \mathbb{I}_m \tag{A.10}$$

where the $ij$th element of $\boldsymbol{G}_{\mathbf{X}}$ is given by $k(\mathbf{X}_i, \mathbf{X}_j)$. For our results in the paper, we use $\epsilon = 0.05$ and a squared exponential kernel for $k$ as in (A.3), setting the length scale $\tau$ automatically to the median pairwise Euclidean distance between rows of $\mathbf{X}$.

The main challenge when using the covariance matrix approach is that it cannot be directly applied to compare layers with different numbers of neurons $n$. To address this, we first convert from $m \times n$ matrices of neural data into a common $m \times p$ size, using the same method as we use for the shape metrics as in (A.4), but without the whitening stage. We can then embed all layers into a common space of $p \times p$ covariance matrices. As in the Gram matrix case, we again run into rank-deficiency issues when $n < m$ (e.g. for the one-hot embedding of targets for which $n = 10$), and so we again regularize by adding a diagonal ridge to the resulting covariance matrices. The full embedding function in the covariance matrix case is given by

$$\mathrm{f}(\mathbf{X}) = \begin{cases} \mathrm{cov}\left([\mathbf{X},\, \mathbf{0}]\right) + \epsilon \mathbb{I}_p & \text{if } n \leq p \\ \mathrm{cov}\left(\mathbf{X}\boldsymbol{U}_{:p}\right) + \epsilon \mathbb{I}_p & \text{if } n > p \end{cases} \tag{A.11}$$

(compare with (A.4)).

Let $\mathbf{P} = \mathrm{f}(\mathbf{X})$ and $\mathbf{Q} = \mathrm{f}(\mathbf{Y})$ be SPD matrices (we are using $\mathbf{P}$ and $\mathbf{Q}$ instead of $\tilde{\mathbf{X}}$ and $\tilde{\mathbf{Y}}$ to use a consistent notation with Pennec (2019)). The AIR metric distance is defined as

$$\mathrm{d}(\mathbf{X}, \mathbf{Y}) = \mathrm{d}_{\mathbb{M}}(\mathbf{P}, \mathbf{Q}) = \sum_i \log(d_i)^2 \tag{A.12}$$

where $d_i$ is the $i$th eigenvalue of $\mathbf{P}^{-\frac{1}{2}}\mathbf{Q}\mathbf{P}^{-\frac{1}{2}}$ (Pennec, 2006; 2019). Since $\mathbf{P}$ is SPD, its singular value decomposition can be written $\mathbf{P} = \boldsymbol{V}\boldsymbol{\Sigma}\boldsymbol{V}^\top$, where $\boldsymbol{\Sigma}$ is a diagonal matrix and $\boldsymbol{V}$ is orthonormal. Following Pennec (2019), we use element-wise square root, exp, and log operations on the singular values to define the matrix square root, matrix exponential, and matrix logarithm:

$$\mathrm{pow}\,(\mathbf{P}, k) = \boldsymbol{V}\,\mathrm{pow}\,(\boldsymbol{\Sigma}, k)\,\boldsymbol{V}^\top$$
$$\exp\,(\mathbf{P}) = \boldsymbol{V}\exp\,(\boldsymbol{\Sigma})\,\boldsymbol{V}^\top$$
$$\log\,(\mathbf{P}) = \boldsymbol{V}\,\mathrm{log}\,(\boldsymbol{\Sigma})\,\boldsymbol{V}^\top$$

where the operations on the left hand side are *matrix* power, exponential, and log, whereas `pow`, `exp`, and `log` operations are performed element-wise on the diagonal of $\boldsymbol{\Sigma}$. $\mathbf{P}^k$ is equivalent to $\mathrm{pow}\,(\mathbf{P}, k)$.

The **geodesics** from $\mathbf{P}$ to $\mathbf{Q}$ is given by

$$\mathrm{geodesic}(\mathbf{P}, \mathbf{Q}, t) = \mathbf{P}^{\frac{1}{2}}\left(\mathbf{P}^{-\frac{1}{2}}\mathbf{Q}\mathbf{P}^{-\frac{1}{2}}\right)^t \mathbf{P}^{\frac{1}{2}} \tag{A.13}$$

(combining equations (3.12) and (3.13) in Pennec (2019)).

**Tangent vectors** in this space are symmetric matrices, and the **logarithmic map** is given by

$$\log_{\mathbf{P}}(\mathbf{Q}) = \mathbf{P}^{\frac{1}{2}}\log\left(\mathbf{P}^{-\frac{1}{2}}\mathbf{Q}\mathbf{P}^{-\frac{1}{2}}\right)\mathbf{P}^{\frac{1}{2}} \tag{A.14}$$

(see equation (3.12) in Pennec (2019)). The **exponential map** of the tangent vector $\boldsymbol{W}$ at $\mathbf{P}$ is given by

$$\exp_{\mathbf{P}}(\boldsymbol{W}) = \mathbf{P}^{\frac{1}{2}}\exp\left(\mathbf{P}^{-\frac{1}{2}}\boldsymbol{W}\mathbf{P}^{-\frac{1}{2}}\right)\mathbf{P}^{\frac{1}{2}} \tag{A.15}$$

(see equation (3.13) in Pennec (2019)). One can easily verify that $\exp_{\mathbf{P}}(\log_{\mathbf{P}}(\mathbf{Q})) = \mathbf{Q}$.

As before, we compute **angles** between triplets of representations $\mathbf{X}, \mathbf{Y}, \mathbf{Z}$ by computing the inner product of the $\log_{\mathbf{Y}}(\mathbf{X})$ and $\log_{\mathbf{Y}}(\mathbf{Z})$ tangent vectors, but unlike the previous metrics the definition of inner products for the AIR metric is not simply the Frobenius inner product. For the AIR metric, the **inner product** of tangent vectors $\boldsymbol{W}$ and $\boldsymbol{V}$ at $\mathbf{P}$ is defined as

$$\langle \boldsymbol{W}, \boldsymbol{V}\rangle_{\mathbf{P}} \equiv \langle \mathbf{P}^{-1}\boldsymbol{W}, \mathbf{P}^{-1}\boldsymbol{V}\rangle_{\mathrm{F}} = \mathrm{Tr}\left(\boldsymbol{W}^\top \mathbf{P}^{-\top}\mathbf{P}^{-1}\boldsymbol{V}\right). \tag{A.16}$$

### A.3.1 Invariances of Affine Invariant Riemannian Metric

We will treat the Gram matrix (A.10) and the covariance matrix (A.11) cases separately. In the Gram matrix case,

- The AIR metric is **shift-invariant**, **scale-invariant**, and/or **rotation-invariant** if and only if the kernel used to compute $\boldsymbol{G}_{\mathbf{X}}$ has the corresponding invariance. Because we use a squared-exponential kernel with a length scale that adapts to the data scale, we have all three invariances.

- The AIR metric is, despite its name, not **affine-invariant** in the sense we are interested in, since affine transformations of $\mathbf{X}$ will in general affect $\boldsymbol{G}_{\mathbf{X}}$ through the nonlinear kernel (e.g. the squared exponential kernel with an *isotropic* length scale is sensitive to non-isotropic scaling of $\mathbf{X}$).

In the covariance matrix case,

- The AIR metric is **shift-invariant** because covariance subtracts the mean.

- The AIR metric is **scale-** and **rotation-invariant** due to the eponymous "affine-invariances" of the metric itself (Pennec, 2006; 2019).

- As in the case of shape metrics discussed above, the AIR metric may or may not be invariant to arbitrary affine transformations of $\mathbf{X}$ due to the restriction to the top $p$ principal components in the embedding stage. Only in the case where $n > p$ and $\boldsymbol{A}$ is a matrix such that $\mathbf{X}\boldsymbol{A}$ changes the subspace of the top $p$ principal components, then the resulting metric is not invariant to $\boldsymbol{A}$.

# B  MODELS AND TRAINING DETAILS

We trained a collection of convolutional networks including both residual networks (He et al., 2016) and VGG (Simonyan and Zisserman, 2014) on CIFAR-10 (Krizhevsky, 2009) using PyTorch (Paszke et al., 2019), managing compute jobs using GNU Parallel (Tange, 2011). We used the OpenLTH framework[6] for training and checkpointing models, using the default hyperparameters for each model.

Following Nguyen et al. (2021), we trained Residual networks of varying widths and depths. The "width" refers to the number of feature channels per convolutional layer, and took on values of $\{16, 32, 64, 128, 160\}$ (corresponding to the base size of 16 multiplied by $\{1\times, 2\times, 4\times, 8\times, 10\times\}$). The "depth" controls the number of residual blocks, according to the formula #blocks $=$ $(\text{depth} - 2)/2$, since each block contains two convolutional layers, and there are two additional preprocessing/projection layers before/after the blocks. We trained models of depths $\{14, 20, 26, 32, 38, 44, 56, 110\}$. The VGG architecture supports "depths" of $\{11, 13, 16, 18\}$, all at the same width. The test accuracy of all models is shown in Table B.2. We analyzed the representational distances and geometry of a subset of these, focusing on depths 14 and 38 (for all widths), and widths 16 and 64 (for all depths).

All models were trained using the default training hyperparameters of OpenLTH[7]. Specifically, all models were trained by stochastic gradient descent for 160 epochs with a batch size of 128 (390.6 batches per epoch of 50k training items), an initial learning rate of 0.1 reducing to 0.01 and 0.001 after 80 and 120 epochs respectively, momentum of 0.9, and weight decay of 0.0001. During training, images were augmented by random horizontal flips and random $\pm 4$ pixel left/right or up/down shifts (padding with zeros).

For the training analysis in Figure 2, we trained a second Resnet-14 with default width (16), and analyzed the representational distance and geometry once every 10 batches for the epoch, since the vast majority of the changes to the network performance and geometry occur in the first epoch.

---

[6]`https://github.com/facebookresearch/open_lth`
[7]`https://github.com/facebookresearch/open_lth`

| Architecture (depth/width) | CIFAR-10 test accuracy (%) |
|:---:|:---:|
| Resnet 14/16 | 90.78 |
| Resnet 14/32 | 92.67 |
| Resnet 14/64 | 93.88 |
| Resnet 14/128 | 94.57 |
| Resnet 14/160 | 94.91 |
| Resnet 20/16 | 91.28 |
| Resnet 20/32 | 93.76 |
| Resnet 20/64 | 94.84 |
| Resnet 20/128 | 95.13 |
| Resnet 20/160 | 95.22 |
| Resnet 26/16 | 91.92 |
| Resnet 26/32 | 93.99 |
| Resnet 26/64 | 94.69 |
| Resnet 26/128 | 95.24 |
| Resnet 26/160 | 95.07 |
| Resnet 32/16 | 93.04 |
| Resnet 32/32 | 94.18 |
| Resnet 32/64 | 94.83 |
| Resnet 32/128 | 95.47 |
| Resnet 32/160 | 94.79 |
| Resnet 38/16 | 92.28 |
| Resnet 38/32 | 94.07 |
| Resnet 38/64 | 95.05 |
| Resnet 38/128 | 94.71 |
| Resnet 38/160 | 94.99 |
| Resnet 44/16 | 93.06 |
| Resnet 44/32 | 94.14 |
| Resnet 44/64 | 95.34 |
| Resnet 44/128 | 94.85 |
| Resnet 44/160 | 94.91 |
| Resnet 56/16 | 93.06 |
| Resnet 56/32 | 94.46 |
| Resnet 56/64 | 94.82 |
| Resnet 56/128 | 95.22 |
| Resnet 56/160 | 95.32 |
| Resnet 110/16 | 93.37 |
| Resnet 110/32 | 94.91 |
| Resnet 110/64 | 94.87 |
| Resnet 110/128 | 95.15 |
| Resnet 110/160 | 95.02 |
| VGG 11 | 91.93 |
| VGG 13 | 93.37 |
| VGG 16 | 93.45 |
| VGG 19 | 93.33 |

Table B.2: Model architectures and performance.

# C Additional figures

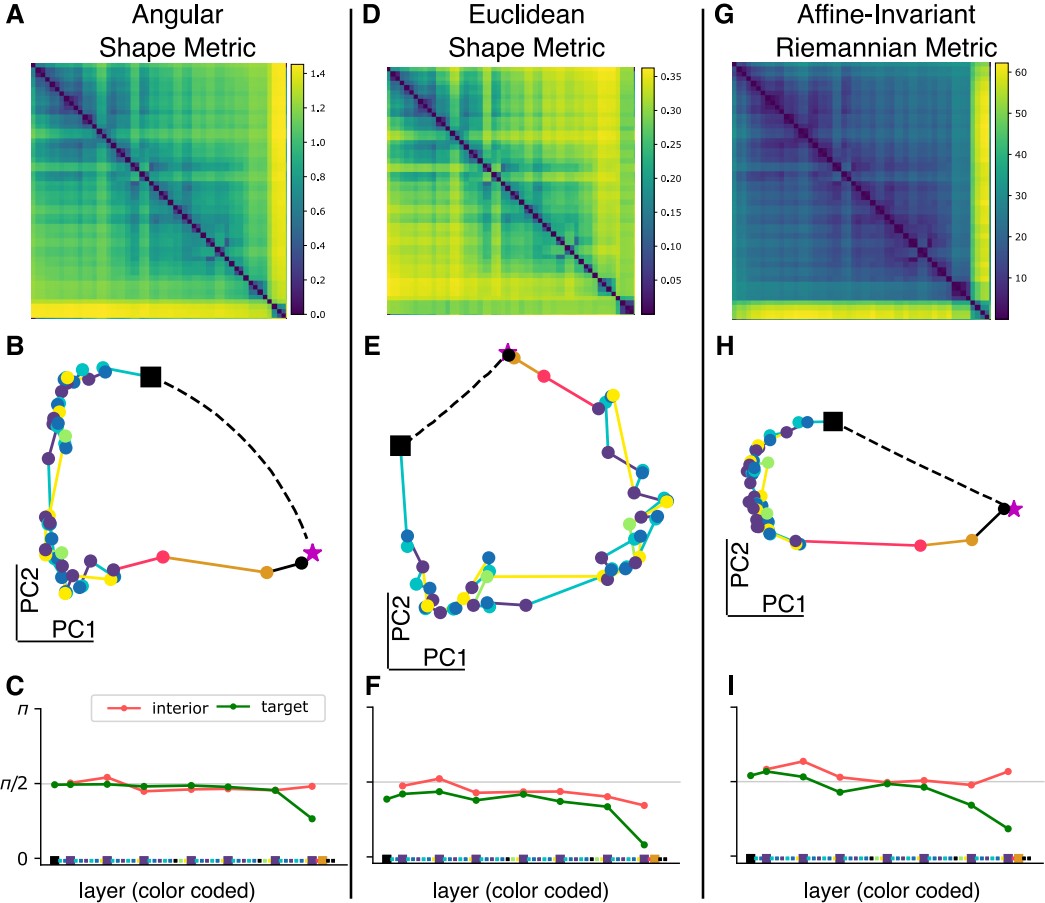

Figure C.1: Comparison of paths taken by the same Resnet-14 model as in Figure 1, but computed with the **Angular Shape Metric** and **Euclidean Shape Metric** of Williams et al. (2021), and the **Affine-Invariant Riemannian Metric** of Shahbazi et al. (2021). **A)** Pairwise distance between layers according to the Angular Shape Metric, computed with $p = 100$ and $\alpha = 0.0$, equivalent to Figure 1E. **B)** 2D embedding of the network's path using MDS, as in Figure 1F. **C)** Internal angles and target angles, as in Figure 1J. **D-F)** Same as A-C but for the Euclidean Shape Metric with $p = 100$ and $\alpha = 0.0$. **G-I)** Same as A-C but for the Affine-Invariant Riemannian Metric using a squared exponential kernel and $\epsilon = 0.1$.

Notably, all metrics agree with the finding that "internal angles" are close to orthogonal throughout the network, and that "target angles" are close to orthogonal for early layers, with the later layers being the only ones that appreciably point in the direction of the targets. Both the Angular Shape Metric and the Affine-Invariant Riemannian metric measure large distances in the last few layers – as the last convolutional layer is projected to logits and then passed through a softmax function. For both of these metrics, the first principal component describing the network path primarily describes just these last few layers (Figure C.1A,B,G,H). This suggests that, while these metrics give sensible results for distances between hidden layers, they may be ill-suited for measuring distances between internal representations and targets.

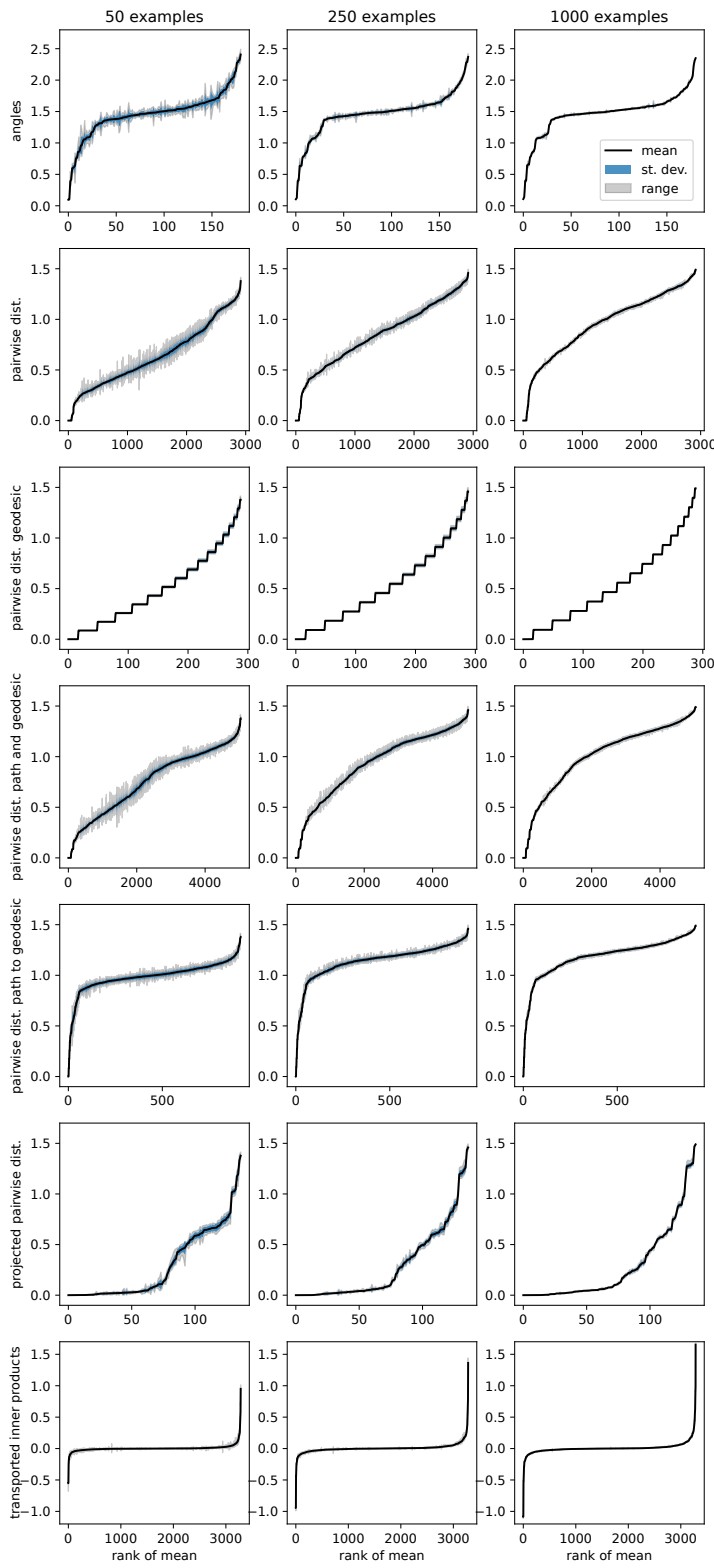

Figure C.2: 10-fold cross validation for quantities computed using the Angular CKA metric on random subsets of CIFAR-10 test examples. Rows correspond to different types of computed quantity, and columns correspond to the number of examples used. Quantities are ordered by the rank of their mean over CV folds (x-axis). Each plot shows the mean (line), standard deviation (blue region), and minimum to maximum range (grey region) per quantity over CV folds (y-axis).

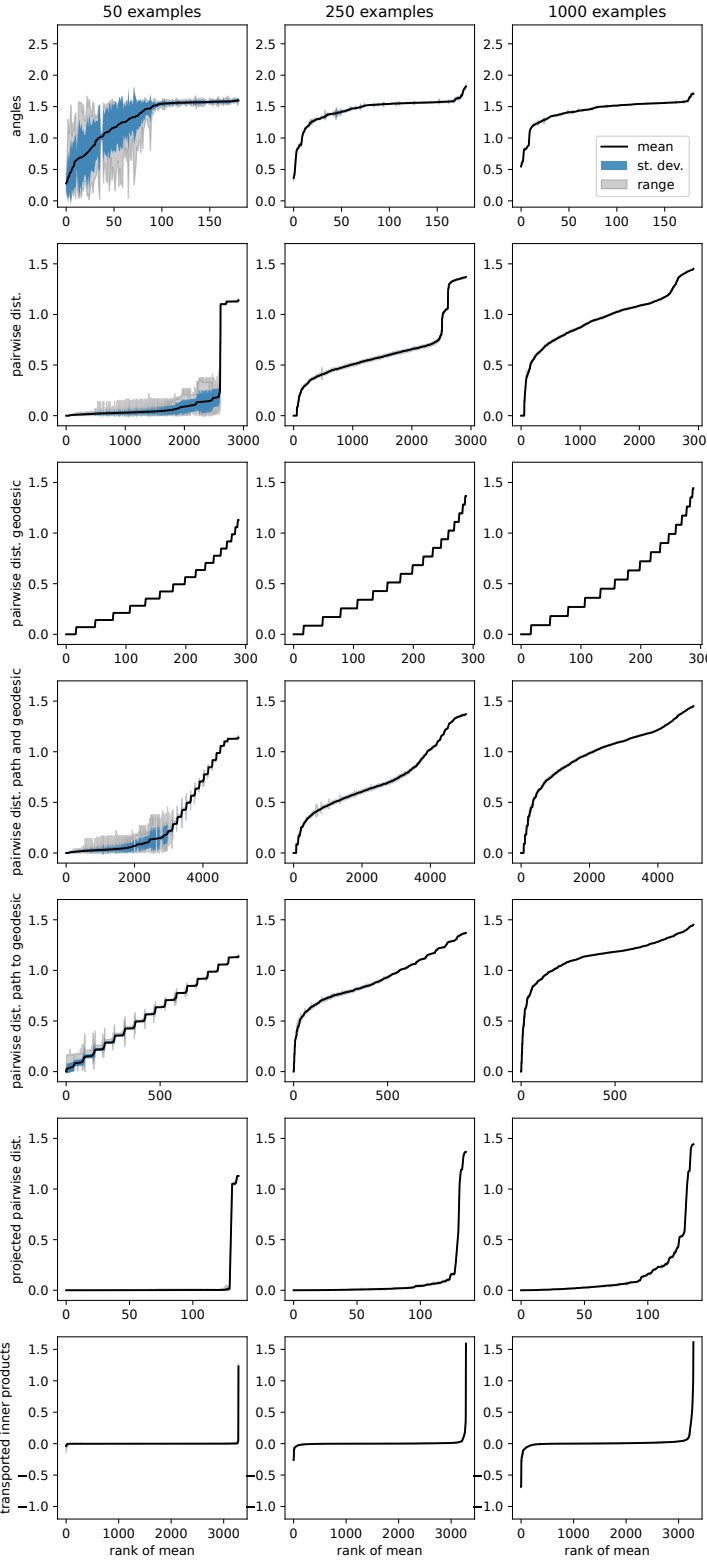

Figure C.3: 10-fold cross validation for quantities computed using the Angular Shape metric on random subsets of CIFAR-10 test examples. Rows correspond to different types of computed quantity, and columns correspond to the number of examples used. Quantities are ordered by the rank of their mean over CV folds (x-axis). Each plot shows the mean (line), standard deviation (blue region), and minimum to maximum range (grey region) per quantity over CV folds (y-axis).

