# OpenReview forum: "Neural Networks as Paths through the Space of Representations"
_ICLR.cc/2023/Conference — Submitted to ICLR 2023_

### Official Review · Reviewer_dSZX · 2022-10-25

**Confidence:** 3
**Correctness:** 3
**Technical Novelty And Significance:** 2
**Empirical Novelty And Significance:** 3
**Recommendation:** 3

**Clarity, Quality, Novelty And Reproducibility:**

There are no issues regarding the clarity or quality of this submission.  I believe there to be no issues regarding reproducibility in this work.  I also believe the proposed approach to be decently novel; whereas they reuse many existing representational metrics, they apply them to the layers within a consistent network and also propose new, additional visualizations about the geometry of the path.

**Strength And Weaknesses:**

A strength of this paper is its proposed geometric visualizations for understanding the internal behavior of neural networks, which is rather novel.  Comparing the paths taken by a wide versus a deep model was also an interesting insight.

A main weakness of this paper is in its inability to explain or hypothesize much of the results it found through its analysis.  Whereas the hypothesis is that training a neural network would cause the representational path to be straight, it happens that they turn more orthogonal instead.  The paths taken by the network also greatly differ from the shortest interpolated path, across all representational metrics.   Such mysteries are not expected or understood, and it is unclear if this is because the metrics are unable to provide a clear picture on the behavior (as the authors mention in the discussion) or if there is something more fundamental about the behavior of a neural network itself.  There is therefore no clean takeaway regarding these proposed metrics - without understanding why the behavior is the way it is (or why it is different from what is expected from these metrics), one might be unable to derive much insights from them and therefore hesitate to use them.

**Summary Of The Paper:**

In this paper, the authors propose thinking about neural network layers as steps in a path that describes how a representation changes steadily towards the target output.  In order to analyze this, they use existing representational distance metrics between the outputs of different layers of the same network.  The authors introduce new visualization methods to understand how a representation changes as it is passed through the network; they propose looking at how the path differs from an shortest-path interpolation of the input and output, projecting onto geodesics, and considering the straightness of the angle that occurs when traveling through two layers.

**Summary Of The Review:**

Overall, I believe this paper is an interesting step towards the direction of understanding how layers interact with each other to compose a complex neural network.  The authors propose such a spatial path analogy, which they set about visualizing layer by layer.  They also propose extra visualizations to understand these paths; interpolating to generate the shortest path, projection, and showcasing the angles between layers.  However, the results that the authors achieved on these proposed visualizations fly against their hypotheses for expected behavior (e.g. the paths do not follow the shortest interpolated paths, nor do the angles suggest a straight path).  It is therefore highly unclear if these visualizations are correct or useful to use, if there are inherent limitations in the representational metrics utilized that inhibit their ability to provide a clear picture into neural network behavior, or some other reason.  Without aligning our hypothesized understanding of how the neural network behaves with the actual behavior according to these visualizations in an interpretable and understandable manner, I believe it would be difficult for the general community to adopt these proposed methods for analysis.  I therefore suggest the authors think deeper about why their results differ greatly from their hypotheses regarding their constructed visualizations, what the current discovered implications are, and if there are other natural metrics that might achieve the expected behavior, before being considered for acceptance.

---

> ### Author Response · Authors · 2022-11-19
> **Minor clarifications**
>
> Thank you for your review! Please see our reply to all above. Regarding your specific comments,
>
> > the results that the authors achieved on these proposed visualizations fly against their hypotheses for expected behavior (e.g. the paths do not follow the shortest interpolated paths, nor do the angles suggest a straight path).
>
> We believe that it is important to state the hypotheses we had *before* running an experiment. It is unfair, in our view, to deduct points from our submission simply because our hypotheses did not align with our results. This is a feature, not a bug, of doing good science!
>
> That being said, we very much sympathize with the reviewer's concern, because we would also like to reach a deeper understanding of our own counter-intuitive results. However, we believe this is a question of scope, and diving deeper into these curious findings will certainly be the subject of our future work. The present paper is primarily about exploration with a newly developed set of tools, as well as creating novel visualizations.
>
> We hope that the reviewer will re-evaluate our paper for what it is (introducing new tools/methods/ideas) rather than for what it isn't (explaining strange phenomena in deep nets) and reconsider their score.
>
> > A strength of this paper is its proposed geometric visualizations for understanding the internal behavior of neural networks, which is rather novel. Comparing the paths taken by a wide versus a deep model was also an interesting insight.
>
> These are among the top messages which we hope readers will glean =)

---

> > ### Comment · Reviewer_dSZX · 2022-12-13
> > **Response**
> >
> > I thank the authors for the response.  I do agree it is important to state the hypothesis before running an experiment, and that it is totally okay to have results that do not align with the hypothesis - yes, this is a feature of doing good science, and surprising results are always welcome.  However, leaving it at that makes the overall work feel rather incomplete!  Some preliminary explanations into why the original hypothesis fails seems almost necessary to “complete” the work and its story.  I understand that this is the planned subject of future work, but I think some aspect of that (even if it is some initial attempts/insight) would really flesh out this submission.
> >
> > I would like to say that this is different from reporting a surprising result of an existing, well-established model or tool.  For example, if someone found a counter-intuitive result with ResNet, this would be worth reporting (and investigating thoroughly), given how prevalent its usage is.  However, in this case, the visualization technique is something proposed by the authors themselves and not yet widely adopted - and therefore any failed hypothesis regarding it is almost entirely the author’s responsibility to uncover and explain, and to convince other members of the community why the failed hypothesis of a *proposed* tool is worth their interest.
> >
> > I feel that, in the extreme, it is difficult to expect the adoption of tools and visualization techniques for which there are no intuitive, sanity-checking demonstrations of it working as expected.  A tool that has strange, unexplained behavior from its genesis seems like the tool itself needs more work before it is ready - and therefore I recommend that the authors continue fleshing out, and understanding their proposed visualizations (in terms of why and how they work).  I therefore maintain my score, and look forward to future iterations of this manuscript from the authors.

---

### Official Review · Reviewer_GUMC · 2022-10-25

**Confidence:** 3
**Correctness:** 3
**Technical Novelty And Significance:** 4
**Empirical Novelty And Significance:** 4
**Recommendation:** 3

**Clarity, Quality, Novelty And Reproducibility:**

The paper is overall well-written.

The analysis is novel to the best of my knowledge.

Code is not provided.

**Strength And Weaknesses:**

[Strengths/Weaknesses]

The most interesting finding to me was that hidden representations in deeper neural networks are closer to the straight line between the input and output representation. While I'm not certain at this point, it is likely that this insight will lead to better generalization guarantees for deeper networks.

The main weakness is that all the experiments are conducted on Cifar-10. I suggest repeating the experiments on at least one more dataset. This dataset has to be sufficiently complex (this excludes MNIST and Fashion MNIST) and sufficiently different from Cifar-10 (this excludes Cifar-100). Some suggestions are STL-10, Caltech, and Mini ImageNet.

Another thing missing in this paper is discussions on the implications of the dissimilarity measures. What does each dissimilarity measure capture and what changes is it invariant to? A nice example is the discussion on CKA by [1] that shows CKA captures alignment between the top singular vectors of the two representations.

I would also like to see what the purpose of the "projection" experiments (Like Fig 1 H-I-J) is and what conclusions one can get from its results.

The discussion section says that the paths are more straight (the angles are closer to pi) at initialization. This is not what I would conclude from the results in Figure 2. Only one dissimilarity measure shows this pattern and even in that subplot there is a lot of variability in the angles. With the other two measures all the angles are close to pi/2.

[Minor comments]

Fig 3 is too crowded and some curves are basically invisible. I suggest breaking it down into multiple figures and moving some to the appendix.

The caption for Fig 4 says that the marker size indicates width. Does this mean that the network width keeps growing all through the legend from top to bottom?

[1] Kornblith, Simon, et al. "Similarity of neural network representations revisited." International Conference on Machine Learning, 2019.

**Summary Of The Paper:**

[Summary]

The submission uses different measures of dissimilarity to characterize the sequence of a neural network's hidden representations at a point in training. They observe that, as the layers progress toward the output layer, the representation gets further from the input representation and closer to the output representation. They also compare th "path" made from this sequence of representations to the shortest path between the input and output representations.

**Summary Of The Review:**

[Decision]

It is hard to evaluate this paper as there is no specified end goal (designing a certain new algorithm or architecture or improving deep learning theory) for this analysis and I cannot determine to what extent the experiment design is appropriate without an ultimate goal. Nevertheless, as long as a study shows clear and reproducible patterns with potential for future work, it is valuable to me. Right now I'm putting a low score since (1) the experiments are on one task and (2) there is no discussion on the implications of the chosen dissimilarity measures but I'll be happy to raise my score if these limitations are addressed unless other reviewers reveal other issues.

---

> ### Author Response · Authors · 2022-11-19
> **Minor clarifications**
>
> Thank you for your review! Please see our reply to all above. Regarding your specific comments,
>
> > The main weakness is that all the experiments are conducted on Cifar-10.
>
> We have new preliminary results on TinyImagenet that initially appear to agree with our original results. Unfortunately, we need a bit more time to check these. We will certainly add these results to future versions of the paper.
>
> > Another thing missing in this paper is discussions on the implications of the dissimilarity measures. What does each dissimilarity measure capture and what changes is it invariant to?
>
> Invariances of the different metrics can be very subtle! For instance, the shape metrics (with whitening) are all invariant to all full-rank affine transformations within the top p principal components of the data, but are not invariant to affine transformations that affect the span of the top p PCs! Meanwhile, the invariances of Angular CKA depend strongly on the choice of kernel used to compute the Gram matrix. We feel that a detailed discussion of these sorts of issues is out of scope for the main text of our paper (especially because, as you note, previous authors have already discussed them at length), but details about invariances for our metrics can be found in the Appendix.
>
> > I would also like to see what the purpose of the "projection" experiments
>
> We are proposing that "projected progress" is a useful visualization tool. Also note that our decomposition of "progress" and "deviation" for each block (Figure 4) relies on the same underlying projection operation, and has a more clear take-home message.

---

> > ### Comment · Reviewer_GUMC · 2022-11-26
> > **Thanks**
> >
> > Thanks for the response and clarifications.
> >
> > I read the other reviews and responses and would like to keep my score as my main concern (lack of diversity in tasks) has not been addressed.
> >
> > I agree with your response to other reviewers that lack of clear motivation or end goal is not a major flaw and the paper is exploring an interesting direction.

---

### Official Review · Reviewer_m1JB · 2022-10-25

**Confidence:** 4
**Correctness:** 3
**Technical Novelty And Significance:** 3
**Empirical Novelty And Significance:** 3
**Recommendation:** 5

**Clarity, Quality, Novelty And Reproducibility:**

From a clarity standpoint I would like to see section 2.3 expanded with explicit references to sections in the Appendix that describe how geodesics, angles, and projections are actually calculated in practice. I think the main text buries too many details.

Can equation (A.10) be found in prior literature on shape space geodesics? If so, could the authors provide a citation. If not could the authros provide a more formal proof that this is indeed a geodesic? What do the geodesics look like for the Euclidean shape metric? Is it true that you only need to fit $\mathbf{R}$ once at the endpoint of the geodesic?

Many expressions in the appendix are asserted without a very clear reference or a self-contained proof. See, for example, equations (A.3) and (A.4). Please flesh this out.

**Strength And Weaknesses:**

The main strength of this paper is to introduce conceptual advances, and the main weaknesses are the empirical demonstrations. While some of the experiments are interesting, I just wish there was "more" in this section of the paper. I also have some concerns about technical details missing from the Appendix (see next section on "clarity").

It is nice that the authors use three different metrics to illustrate their points. I would also be interested in seeing the Euclidean shape metric (not just the angular shape metric) as was introduced in Williams et al. 2021.

I would have loved to see more results on "toy" tasks and models to help build intuition -- e.g. do fully connected networks trained on a simple task like MNIST show similar results to the very deep networks?

I have some reservations about the practice of flattening convolutional layers as the authors describe in footnote 2. Could the authors comment on the alternative procedure in section 2.4 of Williams et al. (2021) for conv layers? I think that paper makes a reasonably compelling case against flattening the conv layers.

I wish the authors were able to dig further into some of their interesting observations. For example, what do the orthogonal jumps at each layer really mean? Can further intuition be developed through targeted experiments?

Finally &mdash; and this is not so much a weakness as a suggestion for future work &mdash; I would ***love*** to see this analysis applied to neural ODEs ([Chen et al. 2018](https://arxiv.org/abs/1806.07366)). The idea is that these will be truly continuous paths through representation space rather than jagged, discrete jumps. I think this may really help shed light on the approximate orthogonal angles observed by the authors.

**Summary Of The Paper:**

The authors leverage recent work (Williams et al., 2021; Shahbazi et al., 2021) that defines metric spaces over neural representations to analyze multi-layer networks as paths through this representational shape space. These metrics were recently developed, and are similar to CKA (Kornblith et al., 2019). The main novelty in this paper comes in section 2.3 where the authors describe how to interpolate along geodesics within this space of representations, measure the angles at two incident geodesic paths, and to perform projection of a point onto a geodesic path. They apply these novel interpretive tools to understand how computations progress in ResNet and VGG architectures. They study how these paths change over training and across "wide" versus "deep" networks. The most interesting result is that the geodesic angles are close to orthogonal at each layer.

**Summary Of The Review:**

I really like the ideas in this paper and I think it could grow into an important form of analysis for the field. Currently I have recommended "borderline reject" due to some of the missing details in the Appendix, and because I am only moderately enthusiastic about the experimental results in this paper (even though I think they are promising). Some of these points are addressable within the rebuttal period and I am very much open to raising my score.

---

> ### Author Response · Authors · 2022-11-19
> **Minor clarifications**
>
> Thank you for your review! Please see our reply to all above. Regarding your specific comments,
>
> > I would also be interested in seeing the Euclidean shape metric
>
> This is indeed available in our python package! We now include a few basic results using the Euclidean shape metric in the Appendix. We opted to focus on Angular CKA in the main text to keep the main paper more focused and to provide more detail.
>
> > I would have loved to see more results on "toy" tasks and models to help build intuition
>
> This is a good suggestion. We have tried our methods on simple fully-connected MNIST models, for instance, and found qualitatively similar results. Unfortunately we do not have the space to add this to our main text, but would be happy to add it to the appendix.
>
> > I have some reservations about the practice of flattening convolutional layers as the authors describe in footnote 2. Could the authors comment on the alternative procedure in section 2.4 of Williams et al. (2021) for conv layers? I think that paper makes a reasonably compelling case against flattening the conv layers.
>
> Thank you for highlighting this important detail. Unfortunately, we are unable to use the (mhw*c) parameterization as we are comparing spatial layers of different sizes, and we are comparing spatial layers with non-spatial layers (such as measuring distance from convolutional layers to targets). The main advantage of the (mhw*c) parameterization shown by Williams et al. is that it converges faster, i.e. for lower m. We now include an analysis in the Appendix (Fig C.3) of signal versus noise while varying m. We found that the value of m=1000 that we use throughout indeed has high SNR. Note that because we are analyzing each layer separately, m*n where n=hwc is effectively smaller than it would be if we concatenated all hidden layers together, as done by Williams et al (to the best of our understanding). This may be why we get good SNR with only m=1000 while Williams et al's results suggested that another order of magnitude larger may be required.
>
> > What do the orthogonal jumps at each layer really mean?
>
> We agree that this is interesting and is something we would like to follow up on in the future. Unfortunately we do not have a completely satisfying answer presently. However, to the extent that you and the other reviewers agree that this result is indeed *counterintuitive*, we hope that you consider it an interesting phenomenon worth sharing.
>
> > I would love to see this analysis applied to neural ODEs
>
> Thank you for the intriguing suggestion! We are excited by the many potential future applications of this work. However, such additions would be out of scope at this stage. That being said, we can share that our preliminary work has found that the path taken by generative models such as a residual normalizing flow [1] follows a significantly different path than the supervised classification models we present in this paper. We will follow up by investigating the reasons for this difference.
>
> > From a clarity standpoint I would like to see section 2.3 expanded
>
> We now give detailed equations for Angular CKA in section 2.3.
>
> > Can equation (A.10) be found in prior literature on shape space geodesics?
>
> Our sources were a combination of [2] and [3].
>
> > What do the geodesics look like for the Euclidean shape metric?
>
> Please see the new Figure C.1
>
> > Many expressions in the appendix are asserted without a very clear reference or a self-contained proof.
>
> We apologize for the lack of clarity here. To be candid, our references were wikipedia, stackexchange, and a bit of back-of-envelope sketching to confirm them. We will be happy to add more formal references, but will need to track them down first.
>
> [1] Chen, R. T., Behrmann, J., Duvenaud, D. K., & Jacobsen, J. H. (2019). Residual flows for invertible generative modeling. Advances in Neural Information Processing Systems, 32.
> [2] Esfandiar Nava-Yazdani, Hans-Christian Hege, Timothy John Sullivan, and Christoph von Tycow-icz. Geodesic analysis in kendall’s shape space with epidemiological applications. Journal of Mathematical Imaging and Vision, 62(4):549–559, 2020
> [3] github.com/geomstats/geomstats

---

> > ### Comment · Reviewer_m1JB · 2022-11-25
> > **Still feeling this to be borderline**
> >
> > I thank the authors for their hard work and see that their new preprint has been extensively revised and edited. Again, I feel enthusiastic about the core concepts and ideas in this paper, but feel that the two "surprising" results are too preliminary to be understood and impactful. My advice remains to expand the experiments section of the paper and to condense the first six pages which describe extensions of the metric space ideas put forth by Shahbazi et al and Williams et al. While I think those conceptual extensions are useful, I think they are over-emphasized at the expense of having more thorough experiments (including toy data or MNIST).
> >
> > I still think that Neural ODEs are the perfect application for this new method / perspective. I understand the authors couldn't fit this in on the short timeframe for this review cycle, but I would be very interested to see if the circuitous path result holds up in that case. The idea is that you could measure things like the instantaneous curvature in shape space since the effective depth of the Neural ODE networks is a continuous variable rather than discretized into layers. My gut tells me that studying this class of models would help the authors understand why they see "orthogonal jumps" from layer to layer in the discrete case.
> >
> > I hope to see this manuscript continue to improve and be published at an ML conference in the near future.

---

### Official Review · Reviewer_kK2H · 2022-11-03

**Confidence:** 4
**Correctness:** 3
**Technical Novelty And Significance:** 3
**Empirical Novelty And Significance:** 2
**Recommendation:** 5

**Clarity, Quality, Novelty And Reproducibility:**

As discussed in the strengths/weaknesses section, I found the figures of this paper pretty hard to read and draw conclusions from. I think the paper presents a novel framing of the networks as moving through a "space", but I'm not sure that's a significant enough contribution.

**Strength And Weaknesses:**

Strengths:
- At a high level, I appreciate that the authors are approaching this from a theory standpoint-- the field of ML is really missing this.
- "representations are not merely separated by some distance, but also by some direction". This is true-- I think this is an interesting reframing and I haven't heard it said like this before.

Weaknesses:
- In general, this paper is interesting, but I'm missing the motivation-- why is it important to trace the paths of the network as it trains? The only answer I could find was in the discussion where the authors discuss the fact that the path is not straight, and maybe we could normalize better to make training more efficient. But the path not being straight isn't that surprising to me, given that gradient descent has no such guarantees.
- The figures are hard to read, and I'm not that convinced that you can draw conclusions from them. For example, "We observe that deep networks have significantly less deviation and slightly less progress per step compared with shallow networks": it would be great to compute some actual correlation coefficients on this.
- Similarly, "wide-model paths and deep-model paths all follow roughly the same trajectory in representational space" and "see salient differences between the VGG and Resnet architectures": there should be numbers to back these up too.
- "This grants us the ability to interpret neural network layers not as discrete jumps from one representation to another, but as a piece-wise smooth curve from inputs to network outputs.". Is this a better mental model? I would think that former might be more accurate-- I am not really sure what the interpolation between two layers really means, given that it's computationally a single step.

Clarity nits:
- nit: Figure 4 is hard to read. Suggestions: make the X axis range smaller, color code resnet_n to be similar colors (how VGG is shades of yellow).
- What is the data on the x axis of E/F/G in Figure 1 (the small squares and dots)?
- in Figure 1, what does "the hypothetical representation at the midpoint of the geodesic between inputs and targets" mean precisely? ie, how did you calculate the pixels in the blue square?
- Figure 3 has too many lines to read. Would recommend separating them out or simplifying.

**Summary Of The Paper:**

This paper looks at the paths (as defined by their representation spaces) taken by models through their training. There is prior work on measuring the distance between two embedding spaces, and the authors expand on this by computing other properties (geodesics, angles, and projections), and show empirical results of comparing ResNet (wide vs deep) and VGG architectures.

**Summary Of The Review:**

The paper presents a novel framing of the networks as moving through a "space", but I'm not sure that's a significant enough contribution.  I also found it generally hard to read and understand, especially the figures, and felt like conclusions were drawn from the figures without enough support.  All in all, I would say reject. That being said, I'm coming from a visualization background, rather than a theory background, so there is a chance that there are other aspects that I couldn't fully evaluate.

---

> ### Author Response · Authors · 2022-11-19
> **Minor clarifications**
>
> Thank you for your thoughtful review and your constructive comments. Please see our reply to all above. Regarding your specific comments,
>
> > why is it important to trace the paths of the network as it trains? ... But the path not being straight isn't that surprising to me, given that gradient descent has no such guarantees.
>
> We may be misunderstanding your comment, but we would like to clarify that we are **not** measuring the path taken by representations over the course of training. We are measuring the path taken "from inputs to outputs" in a single trained network. (Figure 3 shows how the input-output path itself changes over training, but this should not be confused with measuring the path taken by one layer over time).
>
> > This grants us the ability to interpret neural network layers not as discrete jumps from one representation to another, but as a piece-wise smooth curve from inputs to network outputs. Is this a better mental model? I would think that former might be more accurate-- I am not really sure what the interpolation between two layers really means, given that it's computationally a single step.
>
> Thank you for this thoughtful argument. Although the continuous model of computation is not ideal for the neural networks in this work, we believe it is still a valuable perspective for a general-purpose tool that can visualize the paths of arbitrary computations. We are introducing a suite of general-purpose tools for visualizing how representations are transformed, and these tools can in principle be applied to models that have continuous outputs, such as neural ODEs [1] and continuous normalizing flows [2]. Further, the mental model of neural network layers as a discretization of a continuous process is consistent with some other recent work investigating hidden representations [3,4].
>
> > it would be great to compute some actual correlation coefficients on this.
>
> We agree, and we apologize that we have run out of time to add this to our present revision. This will certainly be something we add in the future.
>
> [1] Chen, R. T., Rubanova, Y., Bettencourt, J., & Duvenaud, D. K. (2018). Neural ordinary differential equations. Advances in neural information processing systems, 31.)
> [2] Grathwohl, W., Chen, R. T., Bettencourt, J., Sutskever, I., & Duvenaud, D. (2018). Ffjord: Free-form continuous dynamics for scalable reversible generative models. arXiv preprint arXiv:1810.01367.
> [3] Chan, Kwan Ho Ryan, Yaodong Yu, Chong You, Haozhi Qi, John Wright, and Yi Ma. “Deep Networks from the Principle of Rate Reduction.” arXiv, October 27, 2020. http://arxiv.org/abs/2010.14765.
> [4] He, Hangfeng, and Weijie J. Su. “A Law of Data Separation in Deep Learning.” arXiv, October 30, 2022. http://arxiv.org/abs/2210.17020.

---

### Decision · Program_Chairs · 2023-01-20

**Decision:**

Reject

**Justification For Why Not Higher Score:**

Empirical results are only reported on CIFAR-10 and it is not clear how the findings of the paper is connected to existing phenomena.

**Justification For Why Not Lower Score:**

N/A

**Metareview: Summary, Strengths And Weaknesses:**

In this work, authors use the presentation at different layers of deep networks to visualize how representations evolve across layers of the network. This line of work is interesting and useful. The paper is also well-written and easy to follow. Based on reviewers feedback and my own judgement of the paper, I think the following two are the main weaknesses of the paper:

- All experiments are run on CIFAR-10 dataset. This is a very limiting factor for an empirical paper. I suggest authors to repeat their experiments on more datasets including CIFAR-100, SVHN, ImageNet or even NLP tasks.

- I think authors should go beyond the visualization and observations and suggest how this finding is related to or explaining existing phenomena in deep learning or how it can be potentially used to develop better architectures or training algorithms.

Given the above, I recommend rejection but I suggest authors to resubmit after taking the above points into account.